# Core control principles of the eukaryotic cell cycle

Souradeep Basu[1,3] ✉, Jessica Greenwood[1], Andrew W. Jones[1] & Paul Nurse[1,2]

Cyclin-dependent kinases (CDKs) lie at the heart of eukaryotic cell cycle control, with different cyclin–CDK complexes initiating DNA replication (S-CDKs) and mitosis (M-CDKs)[1,2]. However, the principles on which cyclin–CDK complexes organize the temporal order of cell cycle events are contentious[3]. One model proposes that S-CDKs and M-CDKs are functionally specialized, with substantially different substrate specificities to execute different cell cycle events[4–6]. A second model proposes that S-CDKs and M-CDKs are redundant with each other, with both acting as sources of overall CDK activity[7,8]. In this model, increasing CDK activity, rather than CDK substrate specificity, orders cell cycle events[9,10]. Here we reconcile these two views of core cell cycle control. Using phosphoproteomic assays of in vivo CDK activity in fission yeast, we find that S-CDK and M-CDK substrate specificities are remarkably similar, showing that S-CDKs and M-CDKs are not completely specialized for S phase and mitosis alone. Normally, S-CDK cannot drive mitosis but can do so when protein phosphatase 1 is removed from the centrosome. Thus, increasing S-CDK activity in vivo is sufficient to overcome substrate specificity differences between S-CDK and M-CDK, and allows S-CDK to carry out M-CDK function. Therefore, we unite the two opposing views of cell cycle control, showing that the core cell cycle engine is largely based on a quantitative increase in CDK activity through the cell cycle, combined with minor and surmountable qualitative differences in catalytic specialization of S-CDKs and M-CDKs.

The core eukaryotic cell cycle control system is based on S phase and mitosis being controlled by cyclin-dependent kinases (CDKs) complexed with S phase cyclins (S-CDKs) and mitotic cyclins (M-CDKs), respectively[1,2]. However, there are two fundamentally different views as to how this core CDK system brings about the temporal order of cell cycle events[3]. The first proposes that correct ordering and execution of S phase and mitosis are the consequence of major qualitative differences in the biochemical activities of the S-CDK and M-CDK complexes due to different cyclins[4–6]. These CDK complexes appear sequentially and target different substrates to successively drive S phase and mitosis[5,6]. The second view emphasizes the importance of the total quantitative level of CDK activity in the cell, contributed by both S-CDK and M-CDK, with increasing activity driving the ordering of S phase and mitosis[7,8]. Correct ordering comes about because S phase substrates are phosphorylated at a lower total CDK activity level than mitotic substrates[9].

However, neither of these models is satisfactory in explaining core cell cycle control. If major qualitative differences in CDK complexes bring about cell cycle order, then S-CDKs and M-CDKs should be indispensable as they carry out distinct tasks, but in fact S-CDKs can be deleted in a range of eukaryotes and cell cycle order is still maintained[10–16]. This is consistent with the alternative quantitative hypothesis, but that view predicts that because S-CDKs and M-CDKs provide similar CDK activities they should be interchangeable. However, S-CDKs are unable to fully compensate for M-CDK loss and complete mitosis[17–21].

Here we investigate core cell cycle control in fission yeast and provide a reconciliation of these opposing views, demonstrating the principles on which the eukaryotic cell cycle is organized.

## S-CDK cannot complete mitosis

In the fission yeast, S-CDK consists of Cdk1 (encoded by *cdc2*) complexed with the S phase cyclin Cig2, and M-CDK consists of Cdk1 complexed with the M-cyclin Cdc13. It has been well established that Cdc13–Cdk1 (the fission yeast M-CDK) can compensate for loss of Cig2–Cdk1 (the fission yeast S-CDK)[9], but it is not clear whether S-CDK can compensate for M-CDK. Early studies using a temperature-sensitive *cdc13* mutant showed that Cig2–Cdk1 could initiate but not complete mitosis[17,18], although a more recent study using the temperature-sensitive *cdc13-G282D* strain has suggested that Cig2–Cdk1 can completely compensate for Cdc13–Cdk1 loss[22]. However, the authors of this study cautioned that this temperature-sensitive mutant might not have eliminated all M-CDK activity from the cell, a potential problem because they reported that *cdc13-G282D* cells accumulated dividing septated cells at their restrictive temperature, indicating that mitoses were indeed taking place. To re-examine whether S-CDK is able to overcome loss of M-CDK function, we expressed the S phase cyclin Cig2 in the strong temperature-sensitive mutant *cdc13-9* and observed that, at the *cdc13-9* restrictive temperature, mitosis was completely blocked (Extended

[1]Cell Cycle Laboratory, The Francis Crick Institute, London, UK. [2]Laboratory of Yeast Genetics and Cell Biology, Rockefeller University, New York, NY, USA. [3]Present address: DeepMind, London, UK. ✉e-mail: souradeepb@deepmind.com

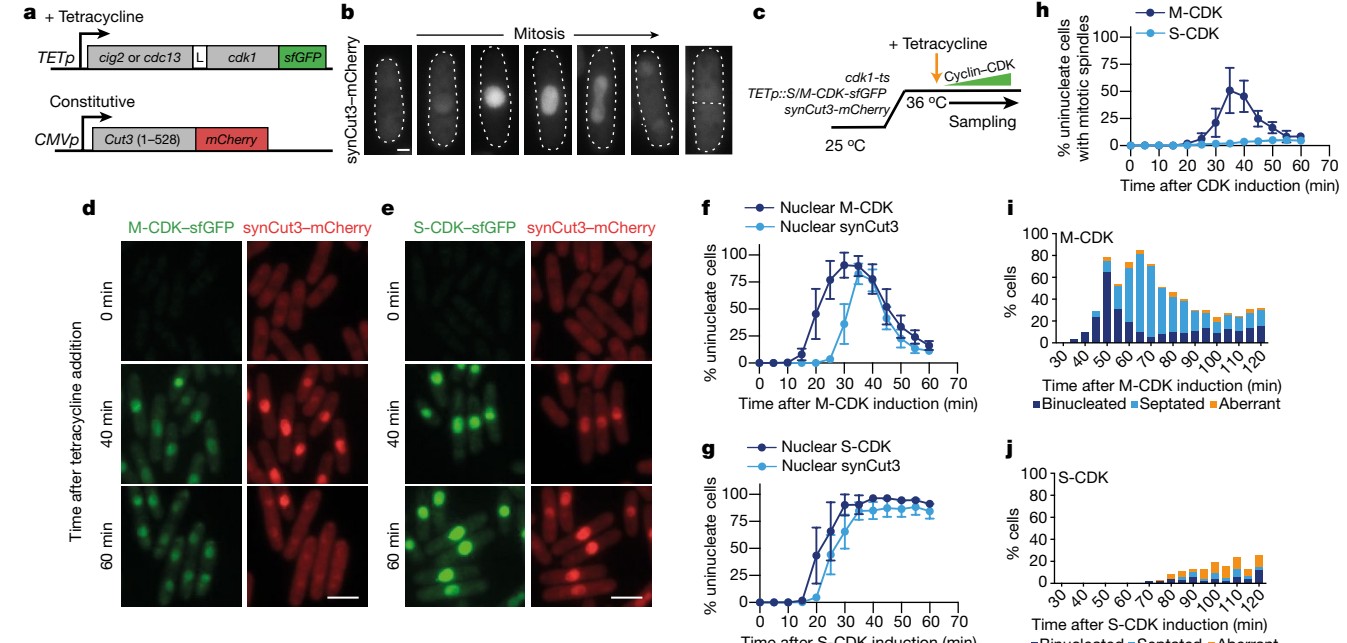

**Fig. 1 | S-CDK can drive mitotic entry but cannot complete mitosis.**
**a**, Schematic of the dual-promoter system. *TETp*, tetracycline-dependent promoter; *CMVp*, constitutive CMV promoter; L, linker peptide region.
**b**, Example images of cells expressing synCut3–mCherry during mitosis. Scale bar, 2 μm. **c**, Experimental schematic for **d**–**j**. **d,e**, Representative images following induction of M-CDK–sfGFP (**d**) or S-CDK–sfGFP (**e**) expression. Scale bars, 10 μm. **f,g**, Percentage of uninucleate cells with nuclear M-CDK–sfGFP (**f**)

or S-CDK–sfGFP (**g**) and nuclear synCut3–mCherry, as a percentage of all cells. *n* = 100 cells per time point, per repeat; *n* = 6 biological repeats. Points, mean; error bars, s.d. **h**, Percentage of uninucleate cells with spindles, as a percentage of all cells. *n* = 100 cells per time point, per repeat; *n* = 3 biological repeats. Points, mean; error bars, s.d. **i,j**, Quantification of mitotic and postmitotic events during longer expression of M-CDK–sfGFP (**i**) or S-CDK–sfGFP (**j**). Data are representative of three biological repeats, with 100 cells per time point.

Data Fig. 1), indicating that S-CDK (Cig2–Cdk1) cannot compensate for M-CDK (Cdc13–Cdk1) in the execution of mitosis.

## S-CDK can trigger mitotic entry

To investigate further the interchangeability of S-CDKs and M-CDKs, we constructed two monomeric S-CDK or M-CDK fusion proteins covalently tagged with Superfolder GFP (sfGFP) under the control of a tetracycline-inducible promoter (Fig. 1a). Endogenous CDK activity was removed using a temperature-sensitive Cdk1 mutant (*cdk1-ts*) in which, at the restrictive temperature of 36 °C, CDK substrate phosphorylation was completely inhibited and cell cycle progression was blocked (Extended Data Fig. 2). We combined this with a mitotic CDK activity biosensor, synCut3–mCherry, which translocates from the cytoplasm into the nucleus at mitosis as a result of direct CDK phosphorylation (Fig. 1a,b)[23].

Endogenous CDK activity was removed by shifting cells to 36 °C for 2 h, and expression of the S-CDK and M-CDK fusion proteins was induced by addition of tetracycline (Fig. 1c). M-CDK and S-CDK were similarly expressed and accumulated in the nucleus of cells (Fig. 1d–g). Expression of both CDKs resulted in import of synCut3 into the nucleus, indicating that sufficient CDK activity was attained to enable mitotic entry (Fig. 1f,g). Following this, M-CDK-expressing cells were able to form spindles, degrade M-CDK, export synCut3–mCherry from the nucleus and undergo nuclear separation (Fig. 1f,h,i). By contrast, cells expressing S-CDK were unable to construct spindles, export synCut3 from the nucleus or degrade the cyclin–CDK complex (Fig. 1g,h) and generated aberrant or incomplete mitotic and cell division events (Fig. 1i,j). The ability of S-CDK and M-CDK to trigger mitotic events was dependent on CDK activity, as kinase-dead S-CDK and M-CDK were unable to cause mitotic events despite high CDK expression levels (Extended Data Fig. 3). We conclude that S-CDK is unable to complete mitosis, but is capable of bringing about the initial stages of mitotic entry.

## Global in vivo cyclin–CDK specificity

That S-CDK cannot substitute for M-CDK for the completion of mitosis and cell division is compatible with a qualitative view of CDK cell cycle control, with S-CDK being unable to phosphorylate essential mitotic substrates. This suggests that there are substrates that are poorly phosphorylated by S-CDK compared with M-CDK in vivo. To investigate this, we developed a time-resolved multiplexed proteomics and phosphoproteomics procedure to monitor both the amount of induced cyclin–CDK present in cells and the ability of that CDK to phosphorylate hundreds of known CDK substrates[4], allowing us to assay the activity of S-CDK and M-CDK in vivo.

We expressed S-CDK and M-CDK fusion protein variants lacking anaphase-promoting complex (APC/C) destruction box motifs (ΔDB) in the absence of endogenous CDK activity. These cyclin–CDK variants were stably expressed and not destroyed by the cyclin destruction machinery (Extended Data Fig. 4). Cells continued to enter mitosis following induction of M-CDK and S-CDK expression over the entire course of the experiment, and thus the entire range of cyclin–CDK levels generated were considered to be physiologically relevant (Extended Data Fig. 4b). The levels of S-CDK and M-CDK were monitored using proteomics, and the phosphorylation status of hundreds of CDK substrates in response to increasing S-CDK or M-CDK levels was quantified using multiplexed phosphoproteomics.

The two complexes were produced similarly after induction (Extended Data Fig. 4c,d), and phosphorylation was normalized to the maximum phosphorylation levels detected in the experiments. In total, 280 previously identified CDK phosphosites were detected, and 276 of these sites were clustered on the basis of their behaviour in response to S-CDK and M-CDK activity (Fig. 2a and Supplementary Table 1). Four distinct phosphorylation behaviours were observed. The largest group of 180 CDK phosphorylation events (cluster 1, ~65%) displayed essentially identical phosphorylation responses to S-CDK

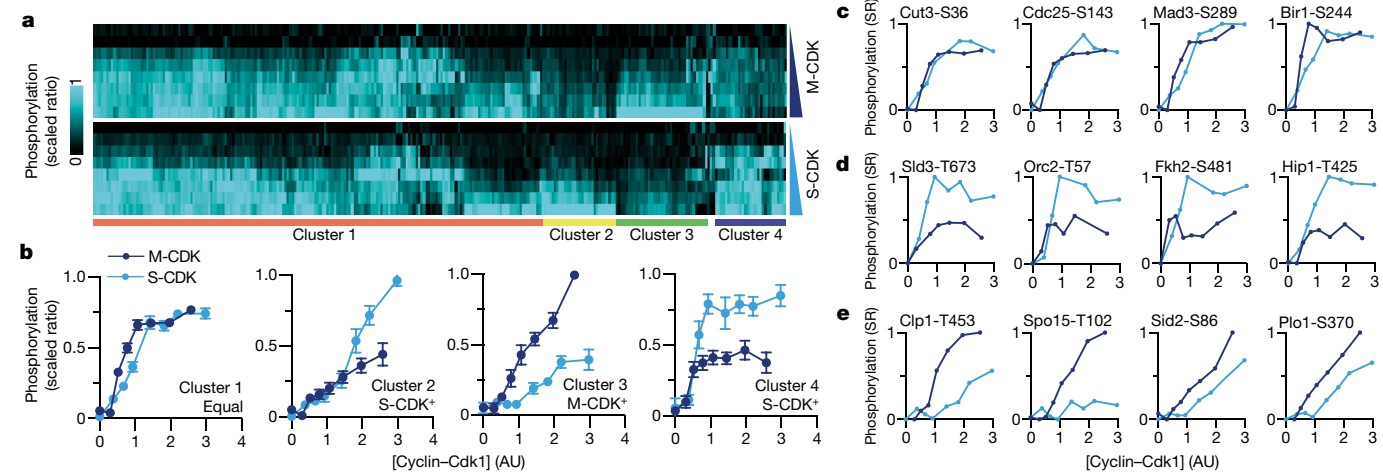

**Fig. 2 | Global phosphorylation by S-CDK and M-CDK complexes.**
**a**, Heatmap of 280 detected CDK phosphorylation events that showed consistent phosphorylation behaviour. Sites are hierarchically clustered into four distinct groups; four sites were not clustered. **b**, Average phosphorylation behaviour for each of the four detected clusters labelled in **a**. A maximum of one aberrant data point per phosphosite was removed prior to representation.

Points, mean; error bars, 95% confidence interval. Cluster 1, $n = 180$ phosphorylation events; cluster 2, $n = 31$ phosphorylation events; cluster 3, $n = 36$ phosphorylation events; cluster 4, $n = 29$ phosphorylation events. AU, arbitrary units. **c**–**e**, Representative substrates from cluster 1 (**c**), clusters 2 and 4 (**d**) and cluster 3 (**e**). Dark blue lines show M-CDK measurements, and light blue lines show S-CDK measurements. SR, scaled ratio.

and M-CDK, reaching identical average maximum phosphorylation levels with similar phosphorylation dynamics (Fig. 2a–c). Clusters 2 and 4 together contained 60 phosphorylation events (~22%) that were preferentially performed by S-CDK, despite most being mitotic substrates (Fig. 2a,b,d). The last of the clustered phosphorylation events (cluster 3, 36 sites, ~13%) included substrates that were better phosphorylated by M-CDK than by S-CDK (Fig. 2a,b,e). However, although these substrates were better phosphorylated by M-CDK, most were still well phosphorylated by S-CDK, with only 11 phosphorylation events on 9 substrates in this cluster (~4% of the total sites) failing to reach 30% phosphorylation downstream of S-CDK at the end of the experiment (Supplementary Table 1).

We conclude that, for the majority of substrates, the preferences of S-CDK and M-CDK are surprisingly similar, suggesting that the S-cyclin and M-cyclin of the cyclin–CDK complex do not impose major differences in CDK substrate specificity in vivo. Thus, the core CDK control system is predominantly reliant on quantitative levels of generic CDK activity contributed by either S-CDK or M-CDK to phosphorylate substrates (Fig. 2b,c), but there are a small number of substrates that rely on qualitative cyclin-specific properties of cyclin–CDK for efficient phosphorylation (Fig. 2b,d,e). The inability of S-CDK to phosphorylate certain substrates within this cluster is probably responsible for S-CDK being unable to substitute for M-CDK. We conclude that core CDK control is hybrid in nature, predominantly quantitative but with low-level qualitative features. The quantitative nature of CDK core cell cycle control may well reflect the situation operative in primaeval eukaryotes 1.0–1.5 billion years ago, which was probably originally based on a single cyclin–CDK complex before gene duplications during subsequent evolution.

## Protein phosphatase 1 restricts S-CDK from executing mitosis

Given the very small differences in CDK substrate phosphorylation between S-CDK and M-CDK, we theorized that S-CDK might be able to execute mitosis if its activity were increased against substrates that it phosphorylates less effectively. To investigate this possibility, we examined the effects of four known inhibitory mechanisms that reduce CDK activity in vivo (Fig. 3a). First, S-cyclins are targeted for degradation by Skp/cullin/F-box (SCF) ubiquitin ligases when complexed with the

F-box adaptors Pop1 and Pop2, or by the APC/C when complexed with Cdh1 (refs. 24,25). Second, CDK activity is inhibited by a CDK inhibitor, Rum1 (ref. 26). Third, interphase CDK activity is opposed by two major phosphatases: protein phosphatase 2A (PP2A) and protein phosphatase 1 (PP1)[27,28]. Finally, CDK is phosphorylated at residues T14 and Y15 by Myt1 and Wee1, which directly inhibits its catalytic activity[29–31].

All of these negative regulators were removed genetically, and their effects on the ability of S-CDK to complete mitosis determined. In the absence of PP1 ($PP1^{dis2}\Delta$), S-CDK-expressing cells could undergo mitosis, but this was not the case with removal of any of the other negative CDK regulators (Fig. 3b and Extended Data Fig. 5). In the absence of PP1, S-CDK-expressing cells constructed spindles and degraded S-CDK coincident with nuclear separation, indicating mitotic exit (Fig. 3c–e), and then proceeded through cytokinesis and cell division (Fig. 3f). Mitotic exit was somewhat delayed compared with mitosis driven by M-CDK, and some aberrant divisions were observed (compare Fig. 3f with Fig. 1i). However, some aberrant divisions were also observed for mitosis driven by M-CDK in the absence of PP1 (Fig. 3g). These experiments demonstrate that PP1 has a major role in restricting S-CDK from executing a full mitosis.

## Centrosomal PP1 restricts S-CDK

PP1 is located throughout the cell but is concentrated at the yeast centrosome (the spindle pole body, or SPB), which organizes the mitotic spindle. Given that cells expressing only S-CDK struggle to construct mitotic spindles, we theorized that PP1 may impose a CDK activity threshold specifically at the SPB, which S-CDK is unable to surpass. If PP1 at the SPB acts as an S-CDK mitotic restriction factor, then removal of PP1 located at the SPB should be sufficient for a S-CDK-mediated mitosis.

PP1 localizes to the SPB through the SPB-localized adaptor protein Cut12, which possesses a bipartite PP1-binding motif (Fig. 4a), and is evicted from the spindle pole at mitosis through phosphorylation of Cut12 by CDK and the NEK kinase Fin1 (ref. 32). We therefore removed PP1 from the centrosome using a mutant allele of *cut12* encoding a variant of Cut12 that is unable to bind PP1 (Cut12$^{\Delta PP1}$; Fig. 4a)[32]. This allele was combined with our in vivo CDK assay system to determine the influence of centrosomal PP1 in restricting S-CDK mitotic activity. Similarly to observations in a wild-type *cut12*+ background, S-CDK

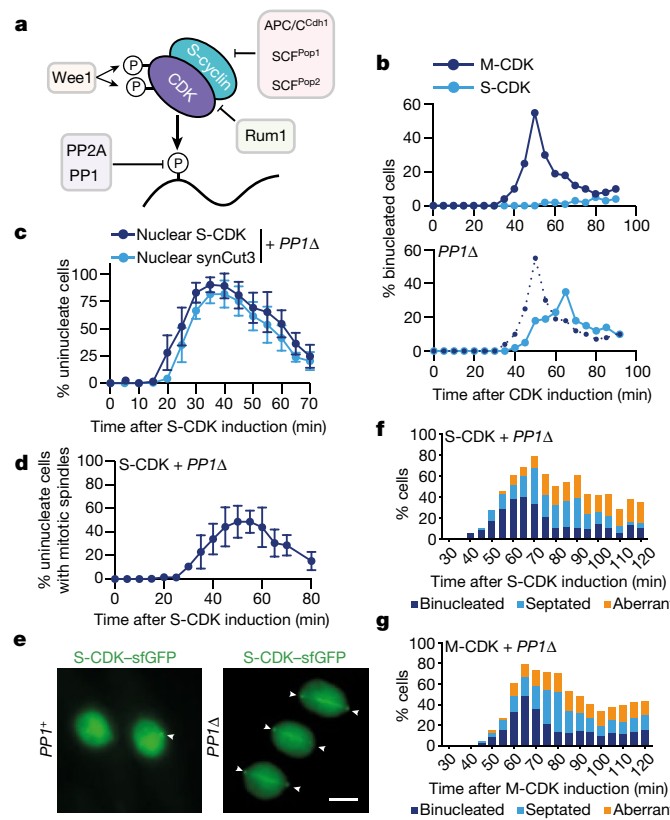

**Fig. 3 | PP1 restricts S-CDK from executing mitosis. a**, Negative regulators of S-CDK and their point of action. **b**, Top, binucleation index after induction of S-CDK–sfGFP and M-CDK–sfGFP expression. Bottom, binucleation index after induction of S-CDK–sfGFP expression in *PP1^dis2*Δ cells lacking PP1 (*PP1*Δ). M-CDK data are replicated from the top panel. Data are representative of two biological repeats, with 100 cells per time point. **c**, Percentage of uninucleate cells with nuclear S-CDK–sfGFP in combination with nuclear synCut3–mCherry, given as a percentage of all cells. *n* = 100 cells per time point, per repeat; *n* = 6 biological repeats. Points, mean; error bars, s.d. **d**, Percentage of uninucleate cells with spindles, given as a percentage of all cells, after induction of S-CDK expression in the absence of PP1. *n* = 100 cells per time point, per repeat; *n* = 3 biological repeats. Points, mean; error bars, s.d. **e**, Example images of mitotic cells in either the *PP1^+* or *PP1^dis2*Δ background. Spindle poles are marked with arrowheads. Scale bar, 3 μm. **f,g**, Quantification of mitotic and postmitotic events during longer expression of S-CDK–sfGFP (**f**) or M-CDK–sfGFP (**g**) in *PP1^dis2*Δ cells from Calcofluor and DAPI staining of fixed cells. Data are representative of three biological repeats, with 100 cells per time point.

accumulated in the nuclei of cells, followed by synCut3 translocation from the cytoplasm into the nucleus (Fig. 4b,c). However, in the absence of SPB-localized PP1, cells were able to undergo mitosis (Fig. 4c, bottom row), construct mitotic spindles (Fig. 4d) and did not exhibit notable numbers of aberrant division events (Fig. 4e). When comparing M-CDK-driven mitosis to S-CDK-driven mitosis in the absence of SPB-localized PP1, there was no difference in the timing of mitosis or in the proportion of cells undergoing division (Fig. 4f). We confirmed this result by removing PP1 from the centrosome by mimicking the phosphorylation events that naturally evict PP1 (ref. [32]) and achieved the same results (Extended Data Fig. 6). These experiments indicate that S-CDK can execute a proper and timely mitosis as long as the negative CDK regulator PP1 is removed from centrosomes.

Next, we used our in vivo kinase assay to determine the changes in CDK substrate phosphorylation when PP1 was removed from the SPB that led to this S-CDK-driven mitosis. As before, the majority of CDK substrates became well phosphorylated by S-CDK, with deletion

of centrosomal PP1 having no impact on their average phosphorylation profile (Fig. 4g). However, there were some differences in substrates that were poorly phosphorylated by S-CDK in wild-type cells (Fig. 4h). The two most prominent of these substrates were the phosphosites on the CDK-counteracting phosphatase Clp1 (Clp1-T453) and the mitotic kinase Plo1 (Plo1-S370), which are both SPB-localized proteins[33,34] (Fig. 4i,j), although additional phosphosites also showed limited increases in phosphorylation following removal of centrosomal PP1 (Extended Data Fig. 7 and Supplementary Table 2). Although removal of PP1 increased the phosphorylation of some S-CDK substrates, other substrates that are poorly phosphorylated by S-CDK were not affected as much by the removal of PP1, demonstrating that removal of centrosomal PP1 does not completely equalize S-CDK and M-CDK substrate specificities (Fig. 4k and Supplementary Table 2).

Although Clp1-T453 and Plo1-S370 phosphorylation was reduced in cells driven by S-CDK (Fig. 2e) and increased to mitotic levels when PP1 was excluded from the SPB (Fig. 4i,j), removal of Clp1, precocious activation of Plo1, or both together was insufficient to allow S-CDK to drive mitosis (Extended Data Fig. 8). However, we found that mimicking a single CDK phosphosite in Cut12 that contributes to centrosomal PP1 eviction (Cut12-T75D) allowed a limited level of mitotic progression (Extended Data Fig. 6a,b). Thus, mechanistically, S-CDK probably fails to evict PP1 from the SPB, which leads to S-CDK being unable to net phosphorylate a subset of mitotic substrates efficiently.

We conclude that some mitotic CDK substrates that are less phosphorylated by S-CDK are restrained from phosphorylation by the presence of PP1 at the centrosome. In the absence of centrosomal PP1, S-CDK is able to phosphorylate substrates essential for mitosis to a mitotically permissible level. These observations support the quantitative view of CDK cell cycle control because, as well as bringing about S phase, S-CDK acting alone can also bring about mitosis if its activity is increased by removal of PP1 located at the centrosome. In further support of this concept, we oscillated S-CDK activity using a non-degradable S-CDK^ΔDB mutant modified such that it could be inhibited by the inhibitor 1-NmPP1. By varying the levels of exogenously added 1-NmPP1, and thus the level of CDK activity, in cells lacking centrosomal PP1 (Fig. 4l), it was possible to drive successive rounds of mitosis and cell division (Fig. 4m). Thus, if PP1 is removed from the centrosome, control of cell division can be brought about simply by varying the quantitative level of S-CDK activity.

## Discussion

We have shown that the in vivo substrate specificities of the major S-CDK and M-CDK in fission yeast are remarkably similar, which is inconsistent with the currently widely accepted qualitative model of core cell cycle control in which these CDKs have markedly different substrate specificities. For 87% of the 276 clustered CDK phosphosites assayed, S-CDK was either equal to or exceeded M-CDK in its ability to phosphorylate CDK substrates, the vast majority of which are mitotic. S-CDK was less effective than M-CDK for only 13% of the phosphosites, and only 4% had activities less than 30% of those seen with M-CDK. This result is not compatible with a purely qualitative view of core CDK cell cycle control. By contrast, our results support a generally quantitative view of core CDK cell cycle control with a small qualitative element.

The small differences in substrate phosphorylation that were observed are probably due to intrinsic dissimilarities between the catalytic properties of S-CDK and M-CDK. However, boosting S-CDK activity by eliminating the CDK-opposing phosphatase PP1 allows phosphorylation of the substrates essential for mitosis. Central to this role of PP1 is its location at the centrosome, as, if PP1 is not allowed to dock to the SPB via Cut12, S-CDK is able to initiate and complete mitosis in a manner identical to M-CDK. Thus, S-CDK is restricted from executing mitosis by restriction of its activity, particularly by PP1 located at the centrosome.

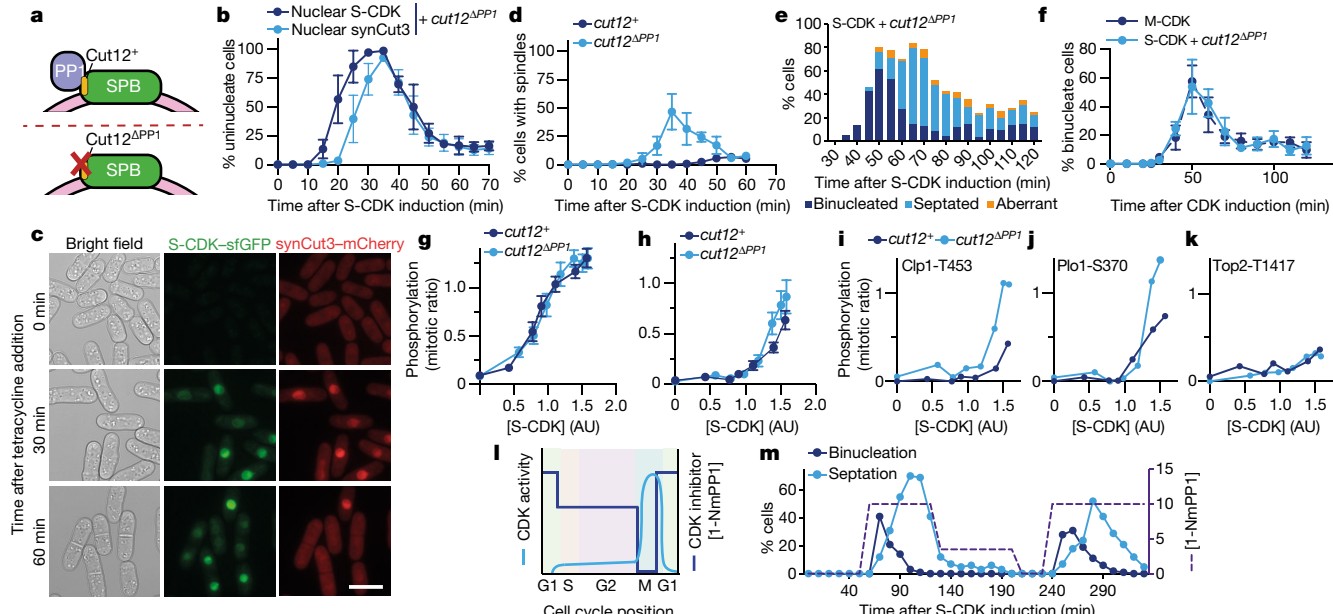

**Fig. 4 | PP1 at the SPB alone restricts S-CDK from executing mitosis. a**, PP1 docking at the SPB through Cut12. Cut12$^{\Delta PP1}$ harbours mutations in the bipartite PP1-binding motif. **b**, Percentage of uninucleate cells with nuclear S-CDK–sfGFP in combination with nuclear synCut3–mCherry, given as the percentage of all cells. $n = 100$ cells per time point, per repeat; $n = 3$ biological repeats. Points, mean; error bars, s.d. **c**, Representative images of cells following induction of S-CDK–sfGFP expression in the cut12$^{\Delta PP1}$ background. Scale bar, 10 μm. **d**, Percentage of uninucleate cells with spindles, as a percentage of all cells, in cells induced to express S-CDK ± Cut12$^{\Delta PP1}$. $n = 100$ cells per time point, per repeat; $n = 3$ biological repeats. Points, mean; error bars, s.d. **e**, Quantification of mitotic and postmitotic events during longer expression of S-CDK with Cut12$^{\Delta PP1}$. Data are representative of two biological repeats, with 100 cells per time point. **f**, Binucleation index after induction of

S-CDK–sfGFP expression in the cut12$^{\Delta PP1}$ background or M-CDK–sfGFP expression in the cut12$^+$ background. $n = 100$ cells per time point, per repeat; $n = 3$ biological repeats. Points, mean; error bars, s.d. **g,h**, Average substrate phosphorylation profiles of the phosphorylated (**g**) and poorly phosphorylated (**h**) substrate clusters by S-CDK in cut12$^+$ or cut12$^{\Delta PP1}$ cells. Points, mean; error bars, 95% confidence interval. $n = 207$ phosphorylation events for **g** and $n = 35$ phosphorylation events for **h**. **i**–**k**, Example substrate phosphorylation profiles of substrates better (**i,j**) or equally (**k**) phosphorylated by S-CDK in the cut12$^{\Delta PP1}$ background. **l**, Schematic of inhibitor-enforced oscillations of CDK activity. **m**, Binucleation and septation indices (left $y$ axis) after induction of S-CDK expression and 1-NmPP1 concentration (right $y$ axis). Data are representative of two biological repeats, with 100 cells per time point.

We conclude that the quantitative view of the core CDK system is the dominating framework that brings about the control and temporal order of S phase and mitosis, with rising overall CDK activity contributed by both S-CDK and M-CDK serving as the core cell cycle organizing principle. However, there are small qualitative differences in substrate specificity between cyclin–CDK complexes that are essential for cell cycle control, which include a PP1-based regulatory process located at mitotic spindle-forming centrosome. This reconciles the two contrasting views of CDK cell cycle control, which we propose is hybrid with a predominantly quantitative nature and a small qualitative influence.

The organism with the best evidence for major qualitative differences in S-CDK and M-CDK substrate specificity is the budding yeast[2]. However, in this system, it has also been shown that a single mitotic cyclin–CDK pair is able to correctly organize and drive S phase and mitosis, but with a delay in cell cycle progression[35]. This indicates that, even in budding yeast, our hybrid view of a quantitative rise in CDK activity coupled with qualitative refining features probably applies. This hybrid view of CDK cell cycle control can be tested directly in other eukaryotes, including in budding yeast and mammalian cells, using the in vivo CDK assay and methodology described here.

Our conclusions are likely to be of relevance to core cell cycle control in other eukaryotes, given the extensive degree of functional redundancy and plasticity for CDKs and cyclins reported in other eukaryotic species[5,11,12,20]. However, the more complex control of the cell cycle required in multicellular eukaryotes, built on interacting tissues and organs, may involve more qualitative regulatory features added to the core eukaryotic cell cycle regulation we describe here.

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

# Methods

## *Schizosaccharomyces pombe* genetics and cell culture

Fission yeast media and growth conditions have been described previously[36]. Strains were constructed either by genetic crossing or by direct transformation as previously described[36]. All strains were checked for correct genotype by colony PCR before use. All strains used and the plasmids used to construct strains are listed in Supplementary Table 3. Unless otherwise stated, all experiments were conducted in yeast extract (YE) medium supplemented with adenine, leucine, histidine and uracil to a final concentration of 0.15 g l$^{-1}$. Cells were grown at 25 °C unless stated otherwise. All experiments were performed with cells in exponential growth, defined as 2.5–10 × 10$^6$ cells per ml. The temperature-sensitive *cdk1* allele used was *cdc2-M26*. To induce expression from the tetracycline-dependent promoter, anhydrotetracycline hydrochloride (Sigma) dissolved in DMSO was added to a final concentration of 0.3125 µg ml$^{-1}$. Vehicle concentration while using 1-NmPP1 or tetracycline was not allowed to exceed 0.1% (vol/vol) of the total culture volume to prevent DMSO-mediated toxicity. To oscillate S-CDK activity in Fig. 4, expression of analogue-sensitive S-CDK$^{ΔDB}$ was initially induced for 50 min. Cells were then washed by filtration into tetracycline-free medium containing 10 µM 1-NmPP1 to allow mitotic progression. Cells were then progressively washed into 3.5 µM 1-NmPP1 and 1-NmPP1-free medium at the time points indicated in Fig. 4 to progress through a whole cell cycle.

## Fluorescence microscopy

All live-cell fluorescence microscopy was performed using a Nikon Ti2 inverted microscope with the Perfect Focus System and Okolab environmental chamber together with a Prime sCMOS camera (Photometrics). The microscope was controlled with Micro-Manager v2.0 software (Open-imaging)[37]. Fluorescence excitation was performed using a SpectraX LED light engine (Lumencor) fitted with standard filters: 470/24 for imaging sfGFP and 575/25 for imaging mCherry; with either, a dual-edge ET-eGFP/mCherry dichroic beamsplitter (Chroma, 59022bs) or a BrightLine quad-edge dichroic beamsplitter (Semrock, FF409-493-573-652) was used. Emission filters were as follows: an ET, EGFP single-band bandpass filter for imaging sfGFP (Chroma, ET525_50m) and a 641/75-nm BrightLine single-band bandpass filter for imaging mCherry (Semrock, FF02_641_75). Images were acquired using a ×100 Plan Apochromat oil-immersion objective (NA, 1.45) at 25 °C. ImageJ software (NIH) was used to measure pixel intensity, adjust brightness and contrast, and render maximum-projection images[38]. Unless otherwise stated, all images represent a single *z* slice across the medial focal plane of cells. For any given figure panel, the same pixel range was applied to all images from the same channel, thus making images in the same channel comparable.

## Determination of cell cycle progression

To score for nuclear division and cell septation indices, 4 µl of cell suspension was heat fixed at 70 °C before addition of DAPI to monitor DNA and Calcofluor to monitor septum formation. For determination of these indices, samples were imaged on a Zeiss Axioskop (×63/1.4-NA objective) or a Nikon Ti2 inverted microscope with the Perfect Focus System (×100/1.45-NA objective; Photometrics). Spindle formation and nuclear enrichment indices were obtained using images from live-cell wide-field imaging. Spindle formation was scored as positive if there was a clear linear trace of sfGFP fluorescence between SPB-like structures. To check for nuclear enrichment of synCut3–mCherry and cyclin–Cdk1–sfGFP, the mean pixel value of a circle encompassing the nucleus was compared to the mean pixel value of a circle of equal area drawn in the cytoplasm. If the nuclear mean value was 1.5 times the cytoplasmic value or greater, this was classed as nuclear enrichment. Scoring for nuclear enrichment indices, cell cycle progression indices and spindle formation indices was conducted on 100 cells per time point.

## Protein extraction and western blotting

Protein was initially extracted from cell cultures by quenching with 100% (wt/vol) ice-cold trichloroacetic acid to a final concentration of 10%. Cells were stored on ice for 20 min, pelleted at 3,000*g* and washed in acetone before storage at −80 °C. After storage, pellets were resuspended in lysis buffer (8 M urea, 50 mM ammonium bicarbonate, 1× cOmplete mini EDTA-free protease inhibitor + 1× phosSTOP phosphatase inhibitor cocktail). Roughly 1.2 ml of acid-washed 0.4-mm glass beads were then added to suspensions, which were subjected to three rounds of beating at 5.5 m s$^{-1}$ for 30 s (FastPrep120). Cell debris was then pelleted at 16,000*g* for 5 min, and supernatant was stored as a whole-cell protein sample at −80 °C. Protein detection by western blotting was performed for Cig2 using a 1:500 dilution of anti-Cig2 (mouse monoclonal) antibody (Abcam, CIG 3A11/5, cat. no. ab10881) blocked with 5% milk in TBS-T. The secondary antibody used was goat anti-mouse (STAR120P, AbD SeroTEC) diluted 1:5,000. Signal was detected using SuperSignal West Femto Maximum Sensitivity Substrate (34095, Life Technologies) and imaged on an Amersham Imager 600.

## Tandem mass tag proteomics

Each protein sample (400 µg) was reduced with 5 mM dithiothreitol (DTT) for 25 min at 56 °C, alkylated with 10 mM iodoacetamide for 30 min at room temperature in the dark and then quenched with 7.5 M DTT. Samples were digested using the SP3 on-bead methodology[39] with the variation that 50 mM HEPES (pH 8.5) was used in place of ammonium bicarbonate. In brief, proteins were bound to the SP3 beads (10:1 beads to protein (wt/wt) ratio) in 50% (vol/vol) ethanol and then washed three times in 80% ethanol, before resuspension in 50 mM HEPES (pH 8.5) with 1:40 (wt/wt) trypsin:protein overnight at 37 °C. The digested samples were arranged in sets of 16 and labelled using the TMTpro 16plex Isobaric Label Reagent Set (Thermo Fisher) according to the manufacturer's instructions. Following labelling and mixing, multiplexed samples were desalted using a C18 SepPak column. Phosphopeptide enrichment was performed by sequential enrichment from metal oxide affinity chromatography (SMOAC, Thermo Fisher) with initial enrichment using the HighSelect TiO$_2$ Phosphopeptide Enrichment kit followed by the HighSelect Fe-NTA Phosphopeptide Enrichment kit (both from Thermo Scientific) for the non-bound flow-through fractions. Phosphopeptides and non-bound flow-through fractions were desalted and fractionated using the High-pH Reversed-Phase Peptide Fractionation kit (Pierce) and analysed on an Orbitrap Fusion Lumos mass spectrometer (Thermo Fisher) coupled to an UltiMate 3000 HPLC system for online liquid chromatography separation. Each run consisted of a 3-h gradient elution from a 75 µm × 50 cm C18 column.

## Mass spectrometry data analysis

MaxQuant (version 1.6.14.0) was used for all data processing. The data were searched against a PomBase[40]-extracted *S. pombe* proteome FASTA file, amended to include common contaminants. Default MaxQuant parameters were used with the following adjustments: phospho(STY) was added as a variable modification (for phosphopeptide-enriched samples), and MaxQuant output files were imported into Perseus (version 1.6.4.7) for further data analysis. The same phosphosite with a different phosphorylation multiplicity was considered to be a separate phosphorylation event. Known CDK sites were excluded if they displayed consistent aberrant phosphorylation behaviour following induction of cyclin–CDK expression. This led to ~9% data loss. Before data representation, up to a single aberrant point per phosphosite was removed for each trace. No individual points were removed for hierarchical clustering. For generation of heatmaps, clustering was conducted using the L1 distance with initial *k*-means clustering (Perseus 1.6.4.7).

## Data representation

All statistical tests were conducted using GraphPad Prism 7 or Prism 8. The central point of all data points gives the mean value, with whiskers delimiting either the 95% confidence interval (for phosphoproteomic data) or s.d. for all other data unless otherwise specified. Where error bars are not present, they are smaller than the size of the data point.

## Reporting summary

Further information on research design is available in the Nature Research Reporting Summary linked to this paper.

## Data availability

All mass spectrometry proteomics data generated have been deposited to the ProteomeXchange Consortium via the PRIDE partner repository with dataset identifier PXD029073.

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

**Acknowledgements** We thank C. Basier for comments on the manuscript and I. Hagan and D. Eckert (Cancer Research UK, Manchester Institute) for *S. pombe* strains carrying mutations in *cut12*. This work was supported by the Francis Crick Institute, which receives its core funding from Cancer Research UK (FC01121), the UK Medical Research Council (FC01121) and the Wellcome Trust (FC01121). In addition, this work was supported by Wellcome Trust grants to P.N. (grant numbers 214183 and 093917), the Breast Cancer Research Foundation (BCRF-21-117), the Lord Leonard and Lady Estelle Wolfson Foundation and the Woosnam Foundation.

**Author contributions** S.B. and P.N. initiated the study. S.B. designed and performed all experiments. S.B. and J.G. generated *S. pombe* strains. A.W.J. performed mass spectrometry and mass spectrometry data analysis. S.B. and J.G. analysed the other data. S.B. and P.N. wrote the manuscript with input from all authors.

**Competing interests** The authors declare no competing interests.

**Additional information**
**Correspondence and requests for materials** should be addressed to Souradeep Basu.

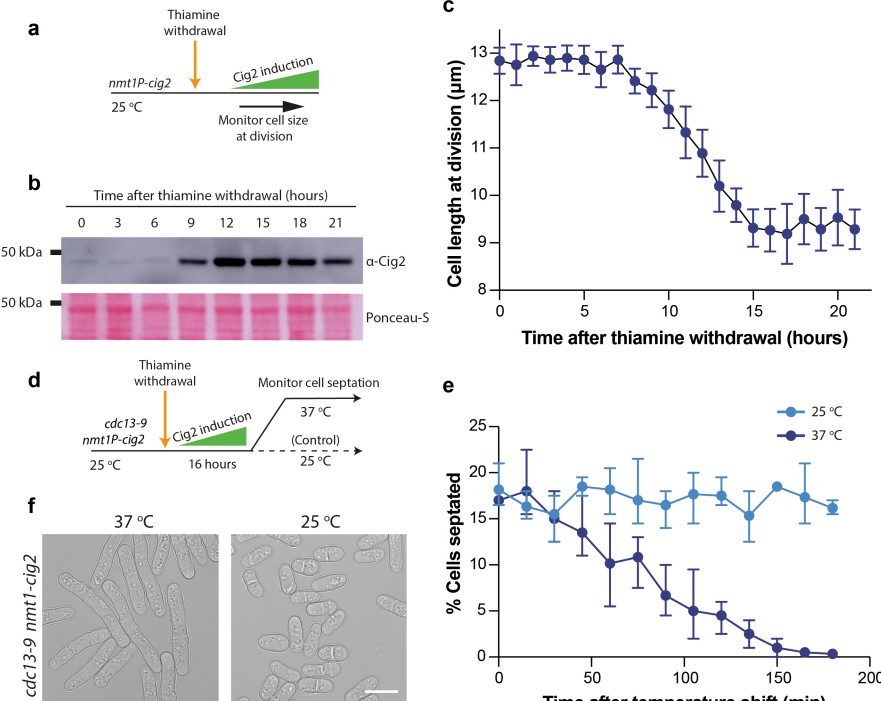

**Extended Data Fig. 1 | Overexpression of Cig2 cannot drive cell division in the absence of Cdc13. a** Experimental outline for panels (b) and (c). 30 μM Thiamine was removed from the growth medium by washing cells into EMM2 via filtration. **b** Western blots for Cig2 (upper) following thiamine withdrawal, with Ponceau-S staining (lower) of the same membrane region as the upper panel given as a loading control. For gel source data, see Supplementary Fig. 1. **c** Cell length at division measurements after thiamine withdrawal by filtration. Cell length at division was determined by visual inspection and measurement of cells with septa. 30 cells were measured per timepoint. n = 3 biological repeats, points shows the mean and error bars delimit SD. **d** Experimental outline for panels (e) and (f). 30 μM Thiamine was removed from media by washing cells into EMM2 via filtration. After 16 h of thiamine withdrawal cells were shifted to 37 °C or held at 25 °C as a control. **e** Percentage of cells with septa as determined by visual inspection. At least 50 cells were counted per time point following shift. n = 3 biological repeats, points show the mean and error bars delimit the range of values. **f** Example images of *cdc13-9 nmt1-cig2* cells taken 19 h after thiamine withdrawal and either shifted to 37 °C after 16 h (left) or maintained at 25 °C (right). Scale bar = 10 μm.

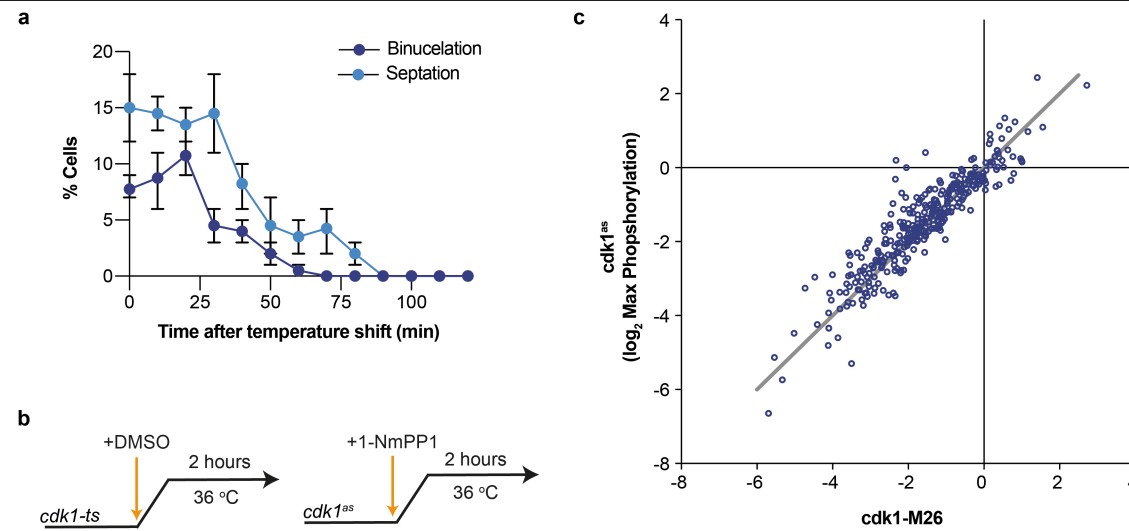

**Extended Data Fig. 2 | Using a cdk1-ts allele to remove endogenous CDK activity. a** Septation and binucleation indices of asynchronous Cdk1-ts (encoded by the *cdc2-M26* allele) cells after shift to the restrictive temperature. 50 cells were counted per timepoint, points represent mean values, and error bars represent range of values over 3 biological repeats. **b** Experimental schematic for compared conditions in panel (c). Samples for phosphoproteomics were taken 2 h after temperature shift or addition of 10 μM 1-NmPP1. **c** Phosphorylation values for 321 previously characterised CDK phosphosites, normalised to mitotic CDK phosphorylation. Sites that did not decrease below zero in both conditions were deemed to be non-genuine CDK phosphosites, and were removed from subsequent analysis. Grey line gives the identity function.

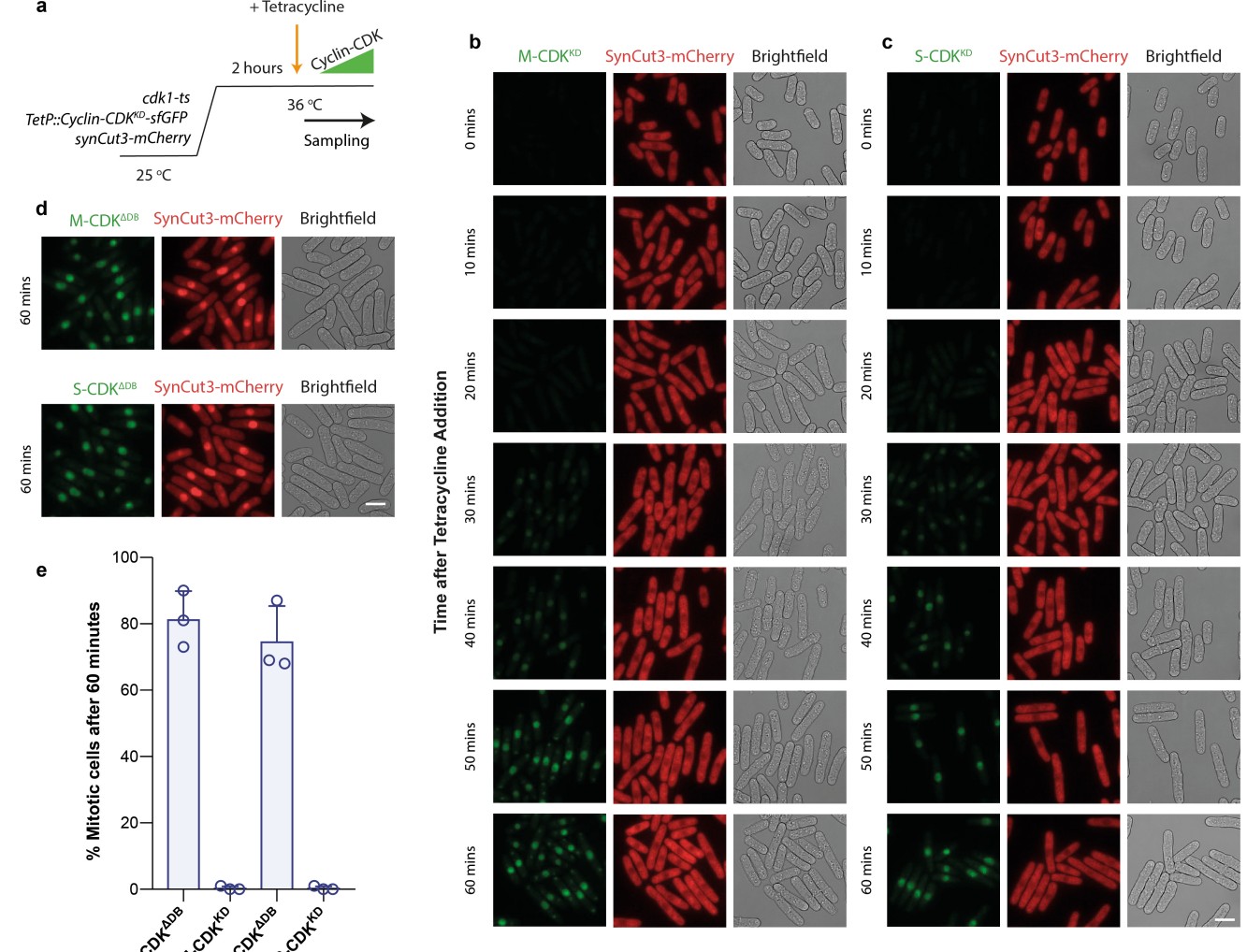

**Extended Data Fig. 3 | Mitotic progression is strictly dependent on induced S-CDK or M-CDK activity. a** Experimental outline for panels (b)-(e). Endogenous CDK activity was removed by shifting Cdk1-ts expressing cells to 36 °C for 2 h. **b, c** Representative images of cells following induction of either M-CDK$^{KD}$-sfGFP (b) or S-CDK$^{KD}$-sfGFP (c). CDK kinase activity is removed through the K33A mutation. Images represent a single Z-slice across the medial focal plane of cells. Images in the same channel are comparable. Scale bar = 10 μm. **d** Representative images of cells following induction of either M-CDK$^{ΔDB}$-sfGFP (top) or S-CDK$^{ΔDB}$-sfGFP (bottom) 60 min after tetracycline addition. Images represent a single Z-slice across the medial focal plane of cells. Images in the same channel are comparable. Scale bar = 10 μm. **e** Quantification of mitotic cells after 60 min of cyclin-CDK induction, as a percentage of all cells. Mitotic cells were classed as cells with nuclear synCut3-mCherry levels 1.5x more than mean cytoplasmic levels. 100/cells per sample, n = 3 samples. Bars give the mean value, error bars give SD.

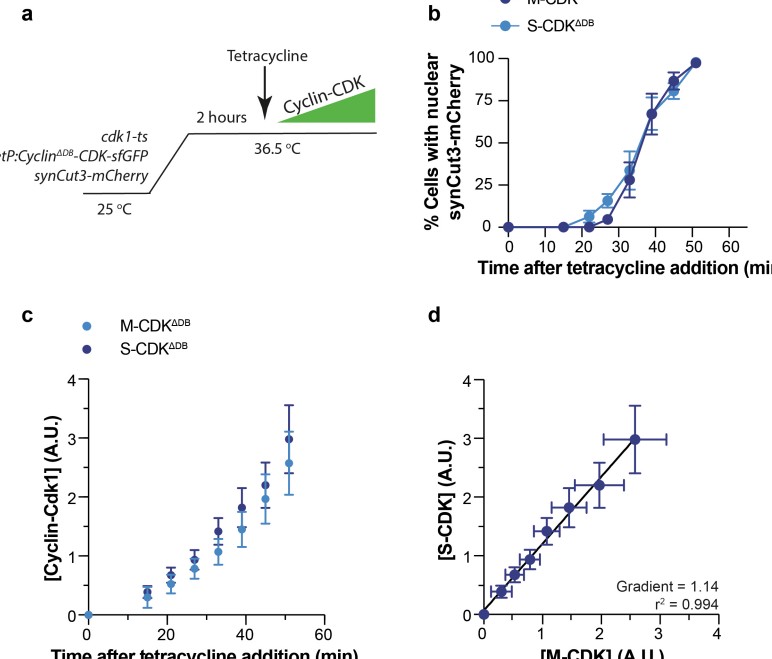

**Extended Data Fig. 4 | Quantification of S-CDK$^{\Delta DB}$ and M-CDK$^{\Delta DB}$ expression. a** Experimental schematic for the induction of truncated cyclin-CDK complexes lacking residues 1-80 of cyclin, which encompass the predicted APC/C recognition degron box sequences of both Cig2 (for S-CDK$^{\Delta DB}$) and Cdc13 (for M-CDK$^{\Delta DB}$). **b** Percentage of uninucleate cells with nuclear synCut3-mCherry after M-CDK$^{\Delta DB}$ and S-CDK$^{\Delta DB}$ induction, as a percentage of all cells. 100 cells per timepoint, per repeat. n = 3 biological repeats. Points give mean, error bars give SD. **c** Proteomic quantification of cyclin-CDK expression after tetracycline induction. n = 15 peptide reporters for M-CDK. n = 12 reporters for S-CDK. Error bars give SEM. **d** S-CDK$^{\Delta DB}$ expression plotted against M-CDK$^{\Delta DB}$ expression. Data taken from panel (c); n = 15 peptide reporters for M-CDK. n = 12 reporters for S-CDK. Error bars give SEM.

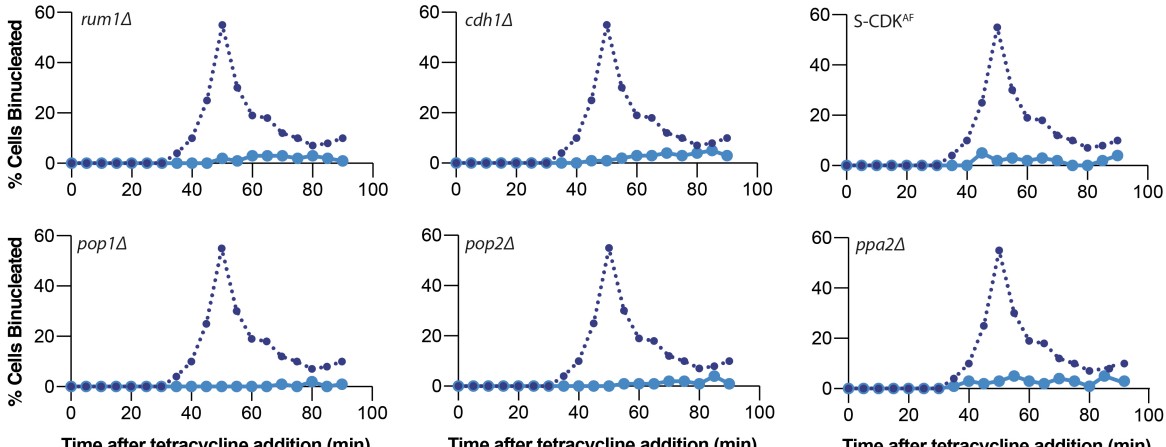

**Extended Data Fig. 5 | Other negative regulators of CDK activity do not restrict S-CDK from executing mitosis.** Binucleation index after induction of S-CDK in the absence of the labelled negative regulators of CDK activity. Instead of the deletion of Wee1, S-CDK$^{AF}$ was expressed carrying unphosphorylatable T14A Y15F mutations on Cdk1, thus mimicking Wee1 kinase loss. Dashed line gives M-CDK induction profile, all reproduced from Fig. 3b.

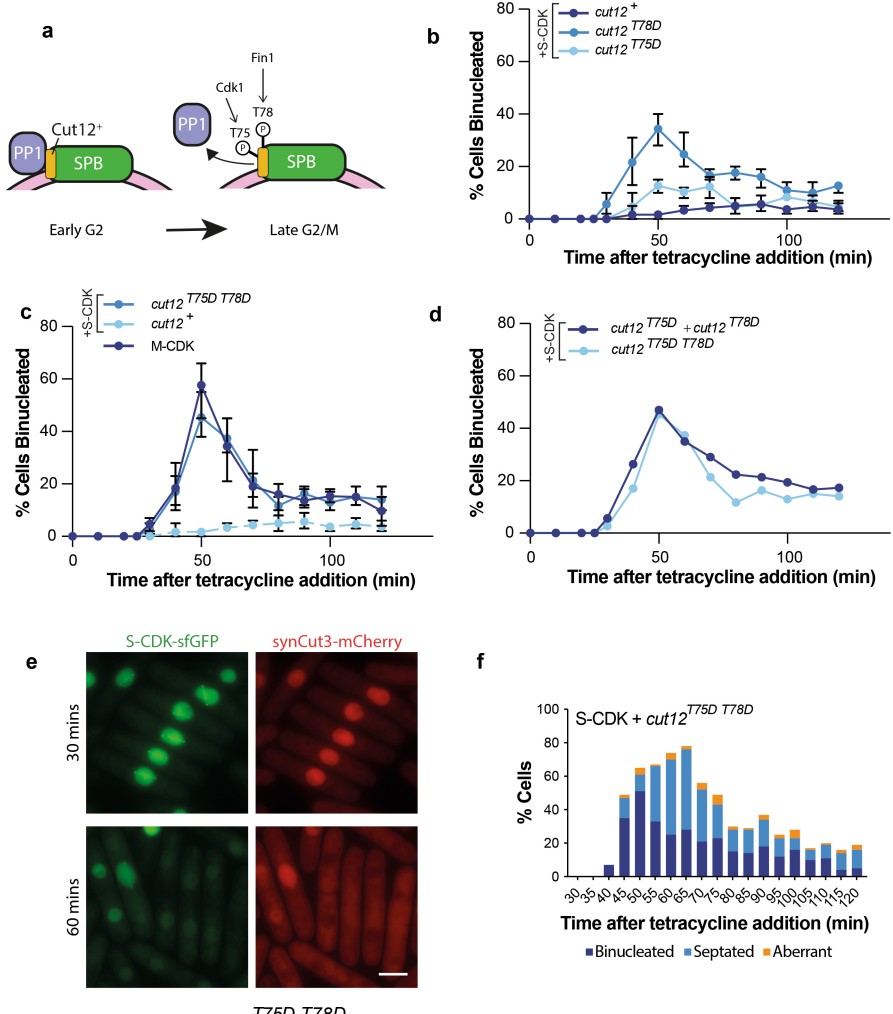

**Extended Data Fig. 6 | Removal of SPB-bound PP1 through mitotic phosphomimetics also allows S-CDK to drive mitosis. a** PP1 regulation through Cut12 phosphorylation by cell cycle kinases. **b** Binucleation index after induction of S-CDK-sfGFP in backgrounds with either wild-type *cut12*[+] or *cut12* mutants with phosphomimetic mutations. Binucleation was measured through DAPI and Calcofluor staining of fixed cells. Septated cells were not included in this measure. 100 cells counted per timepoint. Datapoints represent mean of three biological replicates, and error bars represent SD. **c** As in (b), but including induction of M-CDK in an otherwise wild-type background. Dashed blue line reproduces S-CDK data from panel (b). M-CDK-sfGFP data is

reproduced from Fig. 4f. 100 cells counted per timepoint. Datapoints represent mean of three biological replicates, and error bars represent SD. **d** As in (b), but Cut12[T75D] + Cut12[T78D] curve gives summed mean values from S-CDK inductions in backgrounds with *cut12*[T75D] and *cut12*[T78D] from panel (b). Cut12[T75DT78D] data is directly reproduced from panel (c). **e** Example maximum projection images of S-CDK-sfGFP induced in the presence of the mutant *cut12* allele. Scale bar = 5 μm. **f** Quantitation of mitotic and post-mitotic events during a longer expression of S-CDK induced in the presence of the mutant *cut12*[T75DT78D] allele. 100 cells were counted per timepoint.

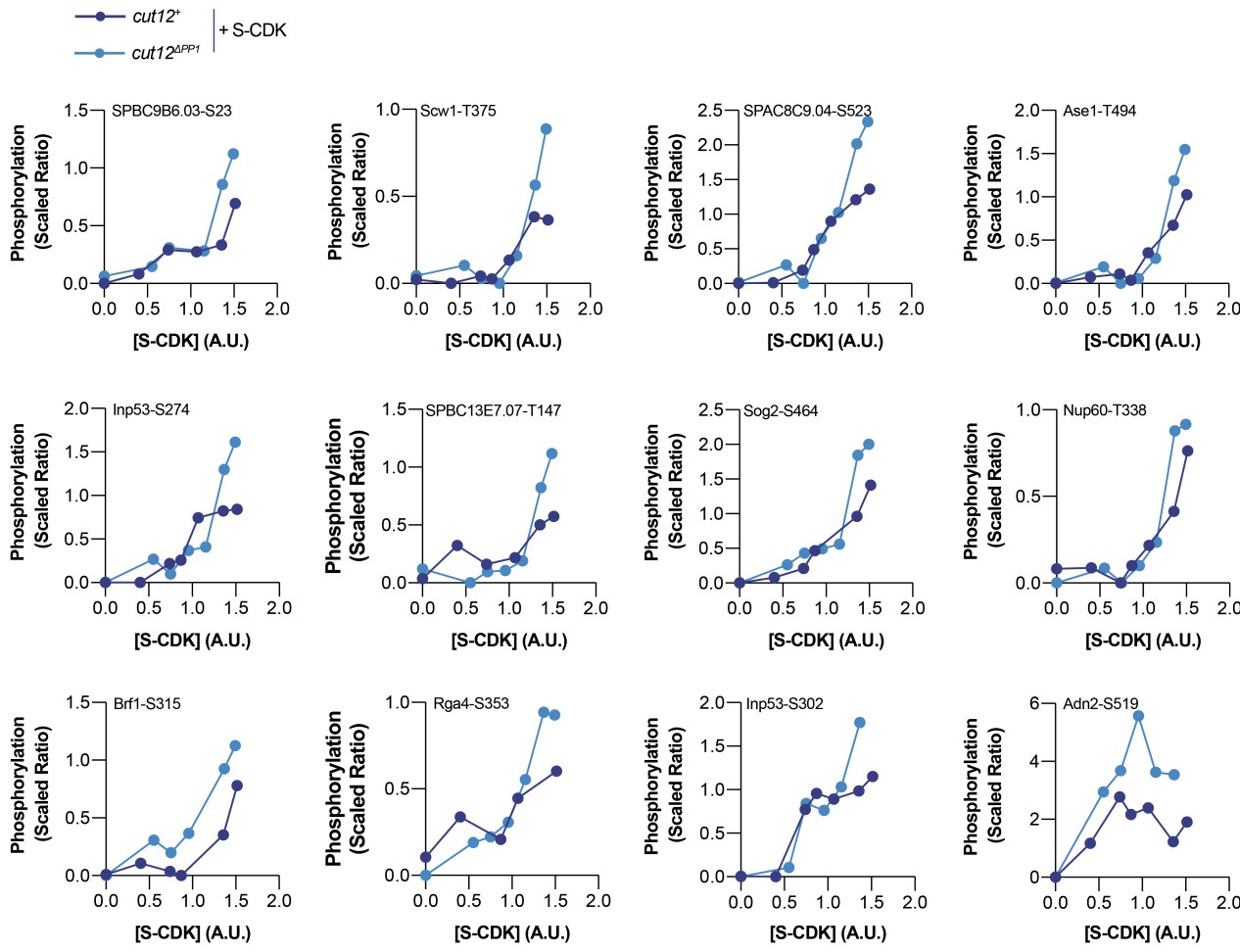

**Extended Data Fig. 7 | Example substrates better phosphorylated by S-CDK in the absence of centrosomal PP1.** Dark blue: S-CDK induction in a wild-type *cut12* background. Light blue: S-CDK induction in a *cut12^{ΔPP1}* background.

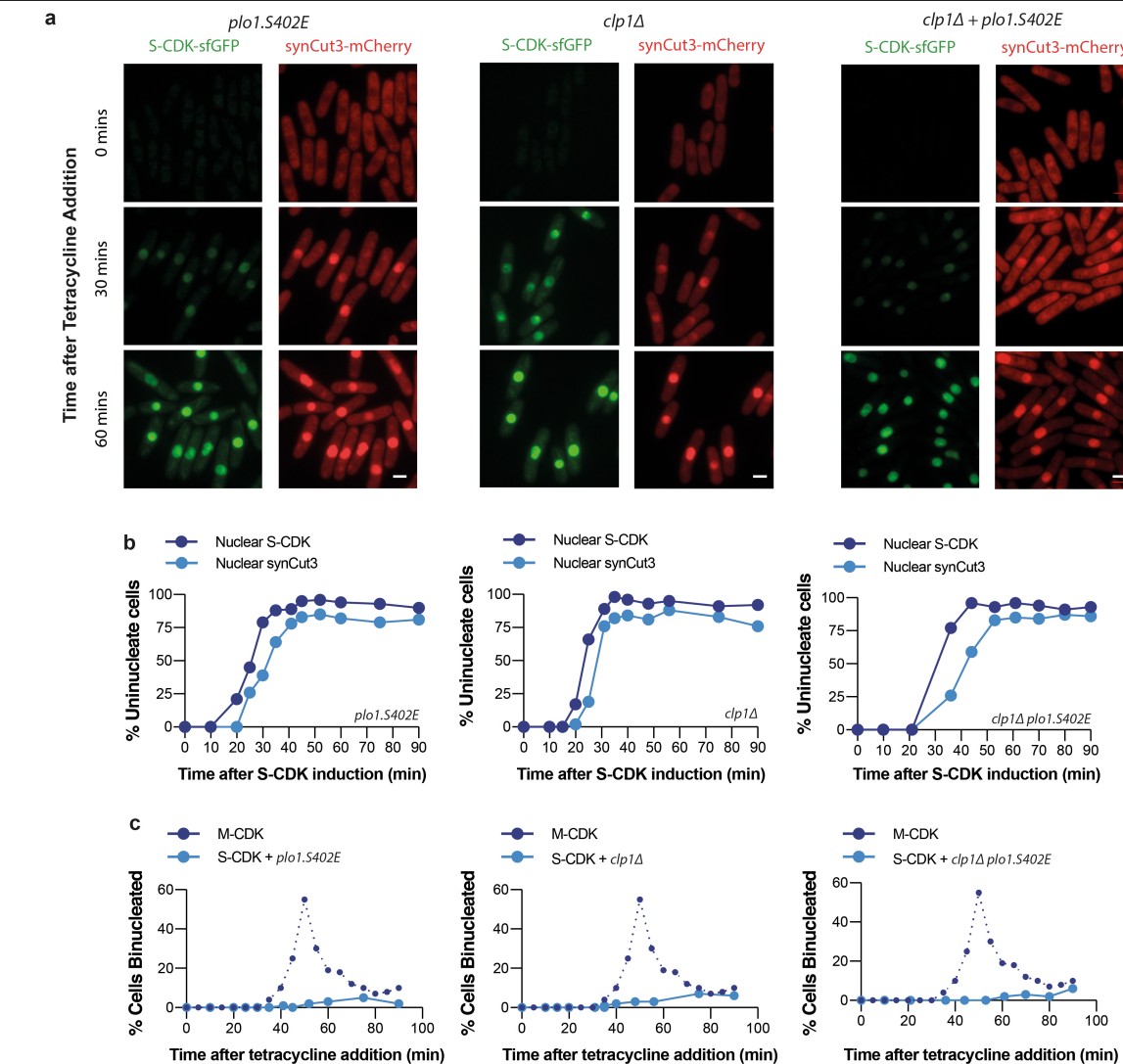

**Extended Data Fig. 8 | Polo kinase and Clp1 phosphatase are not limiting for an S-CDK driven mitosis. a** Example images of cells after S-CDK-sfGFP induction in the annotated background given above images. Scale bar = 5 µm. **b** Percentage of uninucleate cells with nuclear S-CDK-sfGFP and nuclear synCut3-mCherry, as a percentage of all cells. 100 cells per timepoint. **c** Binucleation index after induction of S-CDK in the absence of the labelled regulators of CDK activity. Dashed line gives M-CDK induction profile, all reproduced from Fig. 3b.

| | |
|---|---|

# Reporting Summary

## Statistics

For all statistical analyses, confirm that the following items are present in the figure legend, table legend, main text, or Methods section.

| n/a | Confirmed | |
|---|---|---|
| ☐ | ☒ | The exact sample size (*n*) for each experimental group/condition, given as a discrete number and unit of measurement |
| ☐ | ☒ | A statement on whether measurements were taken from distinct samples or whether the same sample was measured repeatedly |
| ☒ | ☐ | The statistical test(s) used AND whether they are one- or two-sided<br>*Only common tests should be described solely by name; describe more complex techniques in the Methods section.* |
| ☒ | ☐ | A description of all covariates tested |
| ☐ | ☒ | A description of any assumptions or corrections, such as tests of normality and adjustment for multiple comparisons |
| ☐ | ☒ | A full description of the statistical parameters including central tendency (e.g. means) or other basic estimates (e.g. regression coefficient) AND variation (e.g. standard deviation) or associated estimates of uncertainty (e.g. confidence intervals) |
| ☒ | ☐ | For null hypothesis testing, the test statistic (e.g. *F*, *t*, *r*) with confidence intervals, effect sizes, degrees of freedom and *P* value noted<br>*Give P values as exact values whenever suitable.* |
| ☒ | ☐ | For Bayesian analysis, information on the choice of priors and Markov chain Monte Carlo settings |
| ☒ | ☐ | For hierarchical and complex designs, identification of the appropriate level for tests and full reporting of outcomes |
| ☒ | ☐ | Estimates of effect sizes (e.g. Cohen's *d*, Pearson's *r*), indicating how they were calculated |

*Our web collection on statistics for biologists contains articles on many of the points above.*

## Software and code

Policy information about availability of computer code

| | |
|---|---|
| Data collection | Microscopy images were collected using ImageJ 1.52j (NIH). MaxQuant (version 1.6.14.0) was used for mass spectrometry data processing. |
| Data analysis | Microscopy data was analysed using ImageJ 1.52j (NIH). Mass spectrometry data was analysed with Perseus (version 1.6.14.0). GraphPad Prism (Versions 7 and 8) were used for data representation. |

For manuscripts utilizing custom algorithms or software that are central to the research but not yet described in published literature, software must be made available to editors and reviewers. We strongly encourage code deposition in a community repository (e.g. GitHub). See the Nature Portfolio guidelines for submitting code & software for further information.

## Data

Policy information about availability of data

All manuscripts must include a data availability statement. This statement should provide the following information, where applicable:
- Accession codes, unique identifiers, or web links for publicly available datasets
- A description of any restrictions on data availability
- For clinical datasets or third party data, please ensure that the statement adheres to our policy

All mass spectrometry proteomics data generated have been deposited to the ProteomeXchange Consortium via the PRIDE partner repository with the dataset identifier PXD029073.

# Field-specific reporting

Please select the one below that is the best fit for your research. If you are not sure, read the appropriate sections before making your selection.

☒ Life sciences ☐ Behavioural & social sciences ☐ Ecological, evolutionary & environmental sciences

For a reference copy of the document with all sections, see nature.com/documents/nr-reporting-summary-flat.pdf

# Life sciences study design

All studies must disclose on these points even when the disclosure is negative.

| | |
|---|---|
| Sample size | Sample sizes were not predetermined, however we aimed to count 100 cells per time-point to give an accurate reflection of cell population behaviour. |
| Data exclusions | Known CDK sites were excluded from the phosphoproteomic analysis if they: a) displayed consistent aberrant phosphorylation behaviour upon Cyclin-CDK induction, or b) displayed negative phosphorylation values. This led to ~9% data loss. |
| Replication | All attempts at replication were successful, and all experiments were biologically replicated at least twice. |
| Randomization | Randomisation was not relevant to this study, as single genotypes was under study in any given experiment. Therefore, each group only contained a single member. |
| Blinding | Blinding was not possible in this study, as the phenotypes displayed are obvious indicators of the genotype under study. |

# Reporting for specific materials, systems and methods

We require information from authors about some types of materials, experimental systems and methods used in many studies. Here, indicate whether each material, system or method listed is relevant to your study. If you are not sure if a list item applies to your research, read the appropriate section before selecting a response.

## Materials & experimental systems

| n/a | Involved in the study |
|---|---|
| ☐ | ☒ Antibodies |
| ☐ | ☒ Eukaryotic cell lines |
| ☒ | ☐ Palaeontology and archaeology |
| ☒ | ☐ Animals and other organisms |
| ☒ | ☐ Human research participants |
| ☒ | ☐ Clinical data |
| ☒ | ☐ Dual use research of concern |

## Methods

| n/a | Involved in the study |
|---|---|
| ☒ | ☐ ChIP-seq |
| ☒ | ☐ Flow cytometry |
| ☒ | ☐ MRI-based neuroimaging |

# Antibodies

| | |
|---|---|
| Antibodies used | α-Cig2 (mouse monoclonal, primary Ab) (abcam, CIG 3A11/5, Cat# ab10881)<br>α-mouse IgG (goat polyclonal, secondary Ab) (AbD SeroTEC, Cat# STAR120) |
| Validation | Antibody was validated by manufacturer (https://www.abcam.com/cig2-antibody-cig-3a115-ab10881.html) |

# Eukaryotic cell lines

Policy information about cell lines

| | |
|---|---|
| Cell line source(s) | All S. pombe strains used are outlined in Supplementary Table 3. The L972 fission yeast strain background was used in all experiments |
| Authentication | All strains were authenticated by PCR, by positive antibiotic selection, and by negative auxotrophic selection. |
| Mycoplasma contamination | S. pombe cultures do not get contaminated by mycoplasma. |
| Commonly misidentified lines<br>(See ICLAC register) | None. |

