## [Peer Review File · Nature]

Manuscript Title: Core Control Principles of the Eukaryotic Cell Cycle

Reviewer Comments & Author Rebuttals

Reviewer Reports on the Initial Version:

Referees' comments:

Referee #1 (Remarks to the Author):

This exciting, informative and important manuscript from Basu, Jones and Nurse resolves an issue in a long running debate in the cell cycle community as to the importance of specific Cdk-cyclin complexes versus the quantitative provision of activity. This is a natural extension of some exquisite work from Paul Nurse's lab that has shown how the fission yeast cell cycle can be driven by quantitative changes in the activity of a single Cdk-Cyclin complex – a Cdc13-Cdc2 fusion protein. The paper marks a remarkable endorsement of a model proposed in a landmark review with Bodo Stern in 1996 that was not immediately embraced by the community. The data here clearly support the conclusion that the one task where the S phase B type cyclin Cig2 differs from the activity of mitotic cyclin Cdc13 is in the ability of Cdc13 to direct the eviction of PP1 from the SPB. Thus, this work not only unifies the qualitative versus the quantitative model, but it consolidates a lot of observations in a number of systems that the centrosome not only organises the microtubules, but is a key point for integration and execution of cell cycle fate decisions.

I am therefore highly supportive of publication of the manuscript. I do not have any issues with the data or see the need for further experimentation.

Minor points

While the manuscript is very accessible and clearly written, there are a few points of detail that will simply have arisen from too many readings of iterative drafts of a manuscript.

Attention should be paid to how the deviation/error is referred to – for example Fig.1h has SD, Fig.2b “error bars” – is this SE or SD? Figure 4 b and Figure S6B have SEM rather than SD.

The number of experiments e.g. n=3 or n=6 is given for some panels but not others.

There is currently no description of the method used for the experiment of Figure 4m. As this is such a key experiment in the paper it would be useful to provide more detail either in the methods or figure legend.

Attention should be paid to gene/protein naming in Figures 2, 3, 4 and Supplementary Fig. 6.

Line by Line

Line 58 “could initiate mitosis could not complete” > “could initiate mitosis but could not complete”

Line 87 (Figure 1g,h,j) > (Figure 1g,h)

Line 115 and 118 – very minor point, but if the total number of phosphorylation events that can be clustered and hence can be used for calculation as per figure 2b is 276 shouldn't the numbers be 65 and 22 rather than 64 and 21?

Line 119 – (Figure 2a,b,c) > (Figure 2a,b,d)

Lines 148-149 “Finally, CDK is phosphorylated at residues T14 and Y15 by Wee1 and Myt1, which directly inhibit its catalytic” > “Finally, CDK is phosphorylated at residues T14 and Y15 by Myt1 and Wee1, which directly inhibit its catalytic” – Also here unless there is a limit on the number of references that can be listed in the article - additional references are required for the Myt1 phosphorylating T14 – Booher et al. 1997 (JBC 272 22300-22306), Liu et al. 1997 (MCB 17 571-583) and Gallo et al. <https://europepmc.org/article/ppr/ppr309295>

Lines 171-172 “which possesses two PP1 binding motifs” – strictly speaking it is a single bipartite PP1 binding motif. It is phosphorylation between the two component parts that cause its eviction. This is different from the cases, such as the recruitment of PP1 to CenpE where it is phosphorylation within the RVxF motif itself that evicts PP1 (Kim et al. 2010 Cell 142:444).

Line 173 “by CDK and the NEK kinase Sid230” > “by CDK and the NEK kinase Fin130” - it is Fin1 that is the NEK kinase that directly hits T78, it is activated by the NDR kinase Sid2

Line 175 “mutant of Cut12 that lacks PP1 docking motifs (Cut12 Δ PP1, Figure 4a)³⁰. This allele was com” – it is a little confusing here as to whether a protein or a gene allele is being referred to?

Line 177 “wild-type Cut12+ background,” > “wild-type cut12+ background,”

Line 195-196 “Clp1-T452 and Plo1-S370” should be Clp1-T453 and Plo1-S370. No italics as protein modification.

Line 196 “which are both SPB localised proteins” – needs references to papers describing this

Line 220 “280” CDK sites – table 2b lists 276 sites that can be clustered and hence useable for calculation

Line 248 “cyclins reported other eukaryotic” > “cyclins reported in other eukaryotic”

Line 349 “n = 100 cells/time point”. This is confusing. Is it n = 3 as above for h and then 100 cells counted/time point?

Line 382 “following induction of S-CDK-sfGFP” > “following induction of S-CDK-sfGFP in a Cut12 Δ

PP1 background”

Line 552 – please provide concentration of thiamine and means of removal – filtration or centrifugation???? Either here or in methods.

Line 560 – is it EMM or EMM2 – most work from the Nurse lab uses the modified EMM in which potassium hydrogen phallate and the phosphate buffer is switched.

Line 574 – genotype of the analogue sensitive cdc2 mutant – presumably cdc2.asM17 and not cdc2.as1? Also needed in strain list.

Line 616 “Cut12 or Cut12 mutants” >cut12+ or cut12 mutants

Line 624 “with Cut12T75D and Cut12T78D” > “with cut12-T75D and cut12-T78D” “with Cut12T75D and Cut12T78D” because it is the genetic background that is being referred to.

Line 633 “S. pombe strains” > “S. pombe strains and expression plasmids”

Referee #2 (Remarks to the Author):

In this study, Basu et al., address the requirement for fission yeast M-phase cyclin-CDK complex (Cdc13-Cdc2) versus the S-phase cyclin-CDK complex (Cig2-Cdc2) in driving division of yeast *S. pombe*. Two models have been proposed. In a qualitative model, M-CDK and S-CDK phosphorylate distinct subsets of substrates. According to the quantitative model, the level of kinase activity determines substrate specificity. The study of Basu et al., very elegantly reconciles both models.

While it was previously shown that M-CDK can compensate for S-CDK, the authors show that the converse is not true (S-CDK cannot fully replace M-CDK). Specifically, S-CDK cannot complete mitosis, while being capable of initiating mitotic entry. Using unbiased proteomic methods, Basu et al., show that most substrates can be phosphorylated by S-CDK as well as M-CDK. This indicates that the cyclin components of these complexes do not dictate substrate specificity, as assumed by the qualitative model. However, a very small number of phosphorylation events seem to be M-CDK-dependent (i.e., poorly phosphorylated by S-CDK). Importantly, when the activity of S-CDK was increased through genetic removal of PP1, S-CDK kinase was able to complete cell division. Hence, the cell cycle control has both quantitative and qualitative aspects, with the former mechanism being predominant.

The surprising outcome of Basu et al. is the demonstration that in fission yeast centrosomally located PP1 restricts S-CDK activity. When PP1 was genetically removed from the centrosomes, S-CDK was able to complete cell division. The authors conclude that centrosomal PP1 restrains phosphorylation of a small number of substrates by S-CDK1

Overall, this is a very interesting and elegant study. The centrosomal regulation of S-CDK activity is unexpected. My only critique is as follows. Since the authors have identified a very small number of substrates whose phosphorylation is restrained by centrosomal PP1 (Clp1-T452, Plo1-S370), I would

like to see some mechanistic follow-up analyses of these substrates. For example, would phosphomimicking substitutions allow S-CDK to drive cell division?

Referee #3 (Remarks to the Author):

This paper centers on the issue of why S phase occurs before M phase in the eukaryotic cell cycle. The two basic hypotheses are that the different phases of the cell cycle are driven by different cyclin-Cdk complexes with different substrate specificities (the qualitative model) or different amounts of cyclin-Cdk activity (the quantitative model).

Previous work from the Nurse lab (Coudreuse et al. Nature 2010) convincingly argues for the quantitative model in fission yeast--you can run a normal cell cycle with a single Cdk (Cdc2) and cyclin (Cdc13, the M-phase cyclin). This idea was further corroborated through substrate phosphorylation studies in Swaffer et al. Cell 2016. The present studies are sort of the flip side of the coin, focusing on the S-phase cyclin Cig2.

Here Basu et al. show that Cig2 can drive cells into mitosis but not through mitosis, in agreement with refs 17 and 18. They then carry out in vivo dose-response studies with inducible non-degradable Cig2-Cdk and Cdc13-Cdk fusion proteins. Cig2-Cdk still cannot drive cells through mitosis, suggesting that there is something qualitatively different about Cig2 vs. Cdc13 complexes. They then show that most of the phosphopeptides they see by mass spec are phosphorylated similarly by Cig2-Cdk and Cdc13-Cdk, with 21% phosphorylated better by Cig2-Cdk and 13% phosphorylated better by Cdc13-Cdk. Finally, they look at the effects of removing four negative regulators of Cdk on Cig2-Cdk's ability to drive cells through mitosis, and find that in the absence of PP1Dis2, or if PP1 is prevented from localizing to the spindle pole body, Cig2-Cdk becomes able to drive cells through mitosis. This argues that one qualitative difference between Cig2-Cdk and Cdc13-Cdk may be that PP1 protects SPB Cdk substrates better from Cig2-Cdk phosphorylation than from Cdc13-Cdk phosphorylation.

From this they conclude that cell cycle ordering in eukaryotes is mainly established through the Stern and Nurse quantitative model, with some minor qualitative differences between cyclin-Cdk complexes imposed on top of that. They also suggest that the quantitative model was probably the primordial mechanism for ordering S and M phases.

The results are interesting, and they are presented clearly and with appropriate controls. I have reservations though about the interpretations and significance.

To my mind, the Coudreuse work already definitively established that a single cyclin-Cdk complex can drive the whole cell cycle in fission yeast. It seems clear that the quantitative model is a good description of how the fission yeast cell cycle works. Although the authors think the qualitative model is "widely accepted", I do not know anyone who would disagree with the idea that the quantitative model is a good description of what happens in *S. pombe*.

Work on budding yeast has yielded a more nuanced picture. Although there are multiple cyclin-Cdk

complexes activated in temporal waves, they go from least active to most active, in line with the quantitative model (Loog and Morgan Nature 2005; Kõivomägi et al. Mol Cell 2011). But the qualitative model pertains as well, and not just in a minor way. Notably, we now know that Cln3-Cdc28, the first of the cyclin-Cdc28 complexes, has a special ability to phosphorylate the C-terminal domain of Pol II at the initiation of the cell cycle (Kõivomägi et al. Science 2021). Likewise, an S-phase cyclin-Cdk complex (Clb5-Cdc28) phosphorylates some Cdk substrates (~25% of them), including known S-phase regulators, better than an M-phase cyclin-Cdk (Clb2-Cdc28) does (Loog and Morgan Nature 2005). So in budding yeast the general trend from low-to-high activity is there, but different cyclin-Cdk complexes are in fact specialized for different cell cycle events.

In *Xenopus* extracts and mammalian cell lines, it is also clear that different cyclin-Cdk complexes are specialized for different cell cycle events, although the redundancy is greater than most people would have predicted a few decades ago. The Rb and E2F1 proteins are better substrates for cyclin A-Cdks than for cyclin B-Cdks even in vitro, because of cyclin-specific docking interactions (Schulman et al. PNAS 1998). Moreover and may more importantly, M-phase cyclin-Cdks (e.g. cyclin B-Cdk1) are unable to initiate DNA replication because unlike the S-phase cyclin-Cdks (cyclin E-Cdk2, cyclin A-Cdk2), they are excluded from the nucleus during interphase. Targeting cyclin B1-Cdk1 to the nucleus allows it to initiate DNA replication (Moore et al. Science 2003). Thus, there is a qualitative difference between the S-phase and M-phase cyclin-Cdk's localizations that makes the S phase cyclins good and the M phase cyclins bad at initiating S phase, and there are cyclin-specific docking interactions that reinforce that specificity.

So these are my main problems with the work: (1) I was already convinced that the quantitative model is the way to think of what happens in *pombe*; (2) I was already convinced that in budding yeast the quantitative model applies to an extent, but that there are also some major, not minor, specializations of individual cyclins for specific cell cycle events; and (3) It seems clear that in the best-studied animal cell systems, qualitative differences in the cyclin-Cdks contribute in an important way to the ordering of cell cycle events. This is why I am not sure I support publication of this nice paper in Nature.

Minor points:

1. How do the induced Cig2-Cdc2 and Cdc13-Cdc2 activities (Fig 2) compare to the activities present in the natural system?
2. I do not mind speculation (e.g. lines 135-137) but I am not sure it is fair to say that "probably" the quantitative model applied to primordial eukaryotes.
3. It is hard to distinguish between the two blue shades used in the plots in all of the main figures, particularly on the printed page. Perhaps black vs. red would be better.

Referee #4 (Remarks to the Author):

The manuscript by Basu and colleagues utilized *S. pombe* to examine the effects of S-phase CDK and M-phase CDK activity on substrate phosphorylation. This is accomplished through a temperature-sensitive mutant that removes CDK activity at the restrictive temperature (36 degrees). The authors then add back in S-CDK and M-CDK fusion proteins that lack destruction boxes. The levels of the fusion proteins can be controlled by tetracycline. The authors' data demonstrates that most substrates equally phosphorylated by S-CDK or M-CDK.

This reviewer was asked to specifically look at the mass-spectrometry based methods. The authors utilized a single 16plex TMT experiment in Figure 2 to examine the phosphorylation levels of peptides with increasing CDK activities. There are 8 expression (activity) levels for both the S-CDK and the M-CDK fusion proteins ($8 \times 2 = 16$). The experiment setup is fundamentally sound. Since both CDK activities are always measured in the same 16plex there are no missing values in their dataset. The only issue with these data is that there are no replicates for measurements for each activation level—there is just the single 16plex. In theory, this isn't really a problem since there are lot of statistical tests for time-series data that do not use replicates. Overall, the authors cluster their 280 phosphorylation sites into several clusters and then show average membership with error bars. This is a good way to show the data for each cluster.

The approach for the phosphorylation events is a good one with no significant issues within the dataset.

Minor comment:

There are many grammatical and typographical errors throughout the manuscript.

Author Rebuttals to Initial Comments:

We thank the Editor and the Referees for their comments, and the opportunity to revise and improve our work. As a consequence, we have performed additional experimentation and conducted repeat proteomics experiments in order to improve our manuscript, as well as correcting and clarifying the text. We outline our response to the referee's comments below.

Referee #1 (Remarks to the Author):

This exciting, informative and important manuscript from Basu, Jones and Nurse resolves an issue in a long running debate in the cell cycle community as to the importance of specific Cdk-cyclin complexes versus the quantitative provision of activity. This is a natural extension of some exquisite work from Paul Nurse's lab that has shown how the fission yeast cell cycle can be driven by quantitative changes in the activity of a single Cdk-Cyclin complex – a Cdc13-Cdc2 fusion protein. The paper marks a remarkable endorsement of a model proposed in a landmark review with Bodo Stern in 1996 that was not immediately embraced by the community. The data here clearly support the conclusion that the one task where the S phase B type cyclin Cig2 differs from the activity of mitotic cyclin Cdc13 is in the ability of Cdc13 to direct the eviction of PP1 from the SPB. Thus, this work not only unifies the qualitative versus the quantitative model, but it consolidates a lot of observations in a number of systems that the centrosome not only organises the microtubules, but is a key point for integration and execution of cell cycle fate decisions.

I am therefore highly supportive of publication of the manuscript. I do not have any issues with the data or see the need for further experimentation.

Minor points

While the manuscript is very accessible and clearly written, there are a few points of detail that will simply have arisen from too many readings of iterative drafts of a manuscript.

Attention should be paid to how the deviation/error is referred to – for example Fig.1h has SD, Fig.2b “error bars” – is this SE or SD? Figure 4 b and Figure S6B have SEM rather than SD.

We have now clarified and standardised our usage of error bars throughout the text, figures, and legends. Error bars in non-phosphoproteomic data now gives the standard deviation (SD), and error bars in phosphoproteomic data gives the 95% confidence interval (CI) throughout. We now also outline this in the ‘Data representation’ section of the methods [line 556-560].

The number of experiments e.g. n=3 or n=6 is given for some panels but not others.

We now specify the number of experiments aggregated for all figures, or state that a representative example of several biological repeats is shown.

There is currently no description of the method used for the experiment of Figure 4m. As this is such a key experiment in the paper it would be useful to provide more detail either in the methods or figure legend.

Given the limited space in the figure legend, we have outlined the methods for Figure 4m more fully in the methods section [lines 457-462].

Attention should be paid to gene/protein naming in Figures 2, 3, 4 and Supplementary Fig. 6.

These have been corrected, and we have also corrected any errors in gene/protein naming in the main text and methods.

Line by Line

Line 58 “could initiate mitosis could not complete” > “could initiate mitosis but could not complete”

Rephrased and fixed [line 59].

Line 87 (Figure 1g,h,j) > (Figure 1g,h)

Corrected [line 88].

Line 115 and 118 – very minor point, but if the total number of phosphorylation events that can be clustered and hence can be used for calculation as per figure 2b is 276 shouldn't the numbers be 65 and 22 rather than 64 and 21?

We agree, the metric should be on clustered, and not overall sites. This has been corrected [line 118, 121].

Line 119 – (Figure 2a,b,c) > (Figure 2a,b,d)

Fixed [line 122].

Lines 148-149 “Finally, CDK is phosphorylated at residues T14 and Y15 by Wee1 and Myt1, which directly inhibit its catalytic” > Finally, CDK is phosphorylated at residues T14 and Y15 by Myt1 and Wee1, which directly inhibit its catalytic” – Also here unless there is a limit on the number of

references that can be listed in the article - additional references are required for the Myt1 phosphorylating T14 – Booher et al. 1997 (JBC 272 22300-22306), Liu et al. 1997 (MCB 17 571-583) and Gallo et al. <https://europepmc.org/article/ppr/ppr309295>

We have corrected this sentence, and have added the first two peer-reviewed references the Referee has suggested [lines 152-153].

Lines 171-172 “which possesses two PP1 binding motifs” – strictly speaking it is a single bipartite PP1 binding motif. It is phosphorylation between the two component parts that cause its eviction. This is different from the cases, such as the recruitment of PP1 to CenpE where it is phosphorylation within the RVxF motif itself that evicts PP1 (Kim et al. 2010 Cell 142:444).

We have corrected this statement to read ‘which possesses a bipartite PP1 binding motif’ [lines 175-176].

Line 173 “by CDK and the NEK kinase Sid230” > “by CDK and the NEK kinase Fin130” - it is Fin1 that is the NEK kinase that directly hits T78, it is activated by the NDR kinase Sid2

Corrected [line 177].

Line 175 “mutant of Cut12 that lacks PP1 docking motifs (Cut12 Δ PP1, Figure 4a)³⁰. This allele was com” – it is a little confusing here as to whether a protein or a gene allele is being referred to?

Corrected [line 178-179].

Line 177 “wild-type Cut12+ background,” > “wild-type cut12+ background,”

Corrected [line 181].

Line 195-196 “Clp1-T452 and Plo1-S370” should be Clp1-T453 and Plo1-S370. No italics as protein modification.

Corrected [line 199-201].

Line 196 “which are both SPB localised proteins” – needs references to papers describing this

We have now added two references to support this statement [line 201].

Line 220 “280” CDK sites – table 2b lists 276 sites that can be clustered and hence useable for calculation

Corrected [line 235].

Line 248 “cyclins reported other eukaryotic” > “cyclins reported in other eukaryotic”

Corrected [lines 270-272].

Line 349 “n = 100 cells/time point”. This is confusing. Is it n = 3 as above for h and then 100 cells counted/time point?

The referee is correct, n=3 repeats, with 100 cells per timepoint counted in each biological repeat. We have now clarified this, and now use n to refer to the number of repeats of a given experiment throughout the text, rather than the number of cells counted.

Line 382 “following induction of S-CDK-sfGFP” > “following induction of S-CDK-sfGFP in a Cut12 Δ PP1 background”

Corrected [line 423].

Line 552 – please provide concentration of thiamine and means of removal – filtration or centrifugation???? Either here or in methods.

We have now added these details [lines 604-605, 608].

Line 560 – is it EMM or EMM2 – most work from the Nurse lab uses the modified EMM in which potassium hydrogen phallate and the phosphate buffer is switched.

The reviewer is correct, this work was conducted in EMM2, and we have now corrected this throughout.

Line 574 – genotype of the analogue sensitive cdc2 mutant – presumably cdc2.asM17 and not cdc2.as1? Also needed in strain list.

The analogue sensitive variants used in this study are *cdc2.as1*. This is due to *as1* having no hypomorphic phenotype when in the context of the Cyclin-CDK fusion protein¹⁻³. All induced Cyclin-CDKs in this study carry this mutation, and the strain table has been updated to reflect this.

Line 616 “Cut12 or Cut12 mutants” >cut12+ or cut12 mutants

Corrected [line 673].

Line 624 “with Cut12T75D and Cut12T78D” > “with cut12-T75D and cut12-T78D” “with Cut12T75D and Cut12T78D” because it is the genetic background that is being referred to.

Corrected [line 681].

Line 633 “*S. pombe* strains” > “*S. pombe* strains and expression plasmids”

Changed [line 706].

Thus, these issues have all been corrected or clarified, and references added where appropriate. Given the detail and relevance of the comments provided, we especially thank the referee for the care taken in reviewing our manuscript; it is much appreciated, and has improved our paper.

Referee #2 (Remarks to the Author):

In this study, Basu et al., address the requirement for fission yeast M-phase cyclin-CDK complex (Cdc13-Cdc2) versus the S-phase cyclin-CDK complex (Cig2-Cdc2) in driving division of yeast S pombe. Two models have been proposed. In a qualitative model, M-CDK and S-CDK phosphorylate distinct subsets of substrates. According to the quantitative model, the level of kinase activity determines substrate specificity. The study of Basu et al., very elegantly reconciles both models.

While it was previously shown that M-CDK can compensate for S-CDK, the authors show that the converse is not true (S-CDK cannot fully replace M-CDK). Specifically, S-CDK cannot complete mitosis, while being capable of initiating mitotic entry. Using unbiased proteomic methods, Basu et al., show that most substrates can be phosphorylated by S-CDK as well as M-CDK. This indicates that the cyclin components of these complexes do not dictate substrate specificity, as assumed by the qualitative model. However, a very small number of phosphorylation events seem to be M-CDK-dependent (i.e., poorly phosphorylated by S-CDK). Importantly, when the activity of S-CDK was increased through genetic removal of PP1, S-CDK kinase was able to complete cell division. Hence, the cell cycle control has both quantitative and qualitative aspects, with the former mechanism being predominant.

The surprising outcome of Basu et al. is the demonstration that in fission yeast centrosomally located PP1 restricts S-CDK activity. When PP1 was genetically removed from the centrosomes, S-CDK was able to complete cell division. The authors conclude that centrosomal PP1 restrains phosphorylation of a small number of substrates by S-CDK1

Overall, this is a very interesting and elegant study. The centrosomal regulation of S-CDK activity is unexpected. My only critique is as follows. Since the authors have identified a very small number of substrates whose phosphorylation is restrained by centrosomal PP1 (Clp1-T452, Plo1-S370), I would like to see some mechanistic follow-up analyses of these substrates. For example, would phosphomimicking substitutions allow S-CDK to drive cell division?

The referee proposes better mechanistic insight into the nature of S-CDKs inability to execute mitosis, for example by using phosphomimetic mutations. In the revised manuscript, we carried out new experiments aimed at strengthening a mechanistic understanding of the inability of S-CDK to perform mitosis. We considered two mechanistically-focussed hypotheses: (1) S-CDK is unable to target a critical phosphodegron on PP1, causing persistence of PP1 at the SPB at mitosis; (2) S-CDK is unable to activate Polo kinase or is opposed by the CDK-counteracting Cdc14-type phosphatase Clp1 upon the entry into mitosis.

Hypothesis 1

We checked if S-CDK was able to phosphorylate T316 on PP1. This is an evolutionarily conserved phosphosite that results in PP1 degradation at mitosis. It is possible that S-CDK fails to target this site, causing persistence of PP1 at mitosis. However, S-CDK was able to phosphorylate this site to a similar extent to M-CDK, indicating that phosphorylation of this site was not the determining factor between S-CDK and M-CDK (Figure R1).

Figure R1. S-CDK can phosphorylate Dis2 on T316.

Western blot for Dis2 T316 phosphorylation (upper) or Dis2 protein (lower). 1-NmPP1 sensitive Cyclin-CDK complexes were induced for 50 minutes in either the presence 10 μ M of the CDK inhibitor 1-NmPP1 or 0.1% v/v DMSO before cells were fixed.

Hypothesis 2

Given that Plo1-S370 phosphorylation increased upon PP1 removal from the centrosome, we aimed to boost Polo kinase activity by using the phosphomimetic mutation Plo1.S402E, which mimics a phosphorylation event that results in precocious Polo kinase activity at the SPB. We chose this approach as it is not known fully how CDK itself activates Polo kinase, but the S402E mutation is known to promote Polo localisation to the SPB, and its ability to trigger mitotic commitment. However, this was not sufficient to allow S-CDK to execute mitosis (Figure R2).

Figure R2. Precocious activation of Plo1 does not allow S-CDK to execute mitosis

a) Example images of cells after S-CDK-sfGFP induction in the *plo1.S402E* background. Scale bar = 5 μ m.

b) Nuclear enrichment of S-CDK and synCut3-mCherry after S-CDK induction in the *plo1.S402E* background. synCut3-mCherry is used as a readout of mitotic index.

c) Binucleation index after S-CDK induction. Dotted dark blue line gives binucleation index of an M-CDK induction in a *plo1*⁺ background, replicated from Figure 3b in the original text. Light blue gives the S-CDK induction in the *plo1.S402E* background.

Next, given the increase in Clp1-T453 phosphorylation in the absence of centrosomal PP1, we considered to role of the Cdc14-type phosphatase Clp1 in an S-CDK mediated mitosis. S-CDK is unable to trigger anaphase (as judged by persistence of S-CDK protein levels, and CDK activity), and at this pre-anaphase arrest point Clp1 would oppose the CDK phosphorylation of critical substrates. Therefore, we genetically removed *clp1* to try and increase CDK activity prior to anaphase. However, this was also insufficient to allow an S-CDK mediated mitosis (Figure R3).

Figure R3. Removal of Clp1 does not allow S-CDK to execute mitosis.

a) Example images of cells after S-CDK-sfGFP induction in the *clp1Δ* background. Scale bar = 5 μ m.

b) Nuclear enrichment of S-CDK and synCut3-mCherry after S-CDK induction in the *clp1Δ* background. synCut3-mCherry is used as a readout of mitotic index.

c) Binucleation index after S-CDK induction. Dotted dark blue line gives binucleation index of an M-CDK induction in a *clp1⁺* background, replicated from Figure 3b in the original text. Light blue gives the S-CDK induction in the *clp1Δ* background.

Finally, we considered that both Clp1 and Plo1 may be limiting for an S-CDK mediated mitosis. However, S-CDK was unable to drive mitosis when both Clp1 was deleted and Plo1 was precociously activated (Figure R4).

Figure R4. The combination of Polo activation and Clp1 removal does not allow S-CDK to execute mitosis.

a) Example images of cells after S-CDK-sfGFP induction in the *clp1Δ* background. Scale bar = 5 μ m.

b) Nuclear enrichment of S-CDK and synCut3-mCherry after S-CDK induction in the *clp1Δ* background. synCut3-mCherry is used as a readout of mitotic index.

c) Binucleation index after S-CDK induction. Dotted dark blue line gives binucleation index of an M-CDK induction in a *clp1⁺* background, replicated from Figure 3b in the original text. Light blue gives the S-CDK induction in the *clp1Δ* background.

Given that these experiments were not illuminating with respect to strengthening mechanistic understanding, we have not included them as main-text additions, but have included them as an extended data figure (Extended data, figure 8), and make an addition in the text explaining them [lines 209-212].

In proposing phosphomimicking substitution experiments, the referee may also have been thinking of modifying the phosphosites that showed increased phosphorylation upon removal of centrosomal PP1. We realise on re-reading our original manuscript that we may have inadvertently given the impression that there were only two such sites (Clp1-T453 and Plo1-S370 were particularly interesting because of their potential regulatory roles). We have now added data for 12 more similar sites (Extended data, Figure 7; presented here as Figure R5) as well as

highlighting Supplementary table 2 (containing 32 such sites that increase in average and maximum phosphorylation in the absence of centrosomal PP1), which unfortunately makes a comprehensive mechanistic study incorporating all these phosphosite mutants beyond the scope of the present paper [lines 201-203].

The fact that there are number of relevant phosphosites suggests that the failure of S-CDK to execute mitosis is a result of a combination of a number of phosphorylation events. In support of this idea, we originally presented a single CDK-site phosphomimetic on Cut12 (Cut12-T75D) which allows a limited level of S-CDK driven mitotic progression (Extended data figure 6). This mechanistically shows that S-CDK phosphorylation of this site in particular is limiting for mitotic progression, suggesting that S-CDK fails to execute mitosis as it cannot evict PP1 from the centrosome. However, given that Cut12-T75D does not constitute a full rescue suggests there could be a number of other phosphorylation events that are important, including the set we now show in Extended data 7, and others potentially not yet detected by our mass spectrometry studies. We now outline these ideas, and highlight the T75D mutation, in the text on lines 212-216.

Figure R5. Additional substrates showing an increase in phosphorylation by S-CDK in the absence of centrosomal PP1

Light blue: S-CDK induction without centrosomal PP1, in a *cut12^{PP1Δ}* background. Dark blue: S-CDK induction in the presence of centrosomal PP1, in a *cut12⁺* background.

Referee #3 (Remarks to the Author):

This paper centers on the issue of why S phase occurs before M phase in the eukaryotic cell cycle. The two basic hypotheses are that the different phases of the cell cycle are driven by different cyclin-Cdk complexes with different substrate specificities (the qualitative model) or different amounts of cyclin-Cdk activity (the quantitative model).

Previous work from the Nurse lab (Coudreuse et al. Nature 2010) convincingly argues for the quantitative model in fission yeast--you can run a normal cell cycle with a single Cdk (Cdc2) and cyclin (Cdc13, the M-phase cyclin). This idea was further corroborated through substrate phosphorylation studies in Swaffer et al. Cell 2016. The present studies are sort of the flip side of the coin, focusing on the S-phase cyclin Cig2.

Here Basu et al. show that Cig2 can drive cells into mitosis but not through mitosis, in agreement with refs 17 and 18. They then carry out in vivo dose-response studies with inducible non-degradable Cig2-Cdk and Cdc13-Cdk fusion proteins. Cig2-Cdk still cannot drive cells through mitosis, suggesting that there is something qualitatively different about Cig2 vs. Cdc13 complexes. They then show that most of the phosphopeptides they see by mass spec are phosphorylated similarly by Cig2-Cdk and Cdc13-Cdk, with 21% phosphorylated better by Cig2-Cdk and 13% phosphorylated better by Cdc13-Cdk. Finally, they look at the effects of removing four negative regulators of Cdk on Cig2-Cdk's ability to drive cells through mitosis, and find that in the absence of PP1Dis2, or if PP1 is prevented from localizing to the spindle pole body, Cig2-Cdk becomes able to drive cells through mitosis. This argues that one qualitative difference between Cig2-Cdk and Cdc13-Cdk may be that PP1 protects SPB Cdk substrates better from Cig2-Cdk phosphorylation than from Cdc13-Cdk phosphorylation.

From this they conclude that cell cycle ordering in eukaryotes is mainly established through the Stern and Nurse quantitative model, with some minor qualitative differences between cyclin-Cdk complexes imposed on top of that. They also suggest that the quantitative model was probably the primordial mechanism for ordering S and M phases.

The results are interesting, and they are presented clearly and with appropriate controls. I have reservations though about the interpretations and significance.

To my mind, the Coudreuse work already definitively established that a single cyclin-Cdk complex can drive the whole cell cycle in fission yeast. It seems clear that the quantitative model is a good description of how the fission yeast cell cycle works. Although the authors think the qualitative model

is “widely accepted”, I do not know anyone who would disagree with the idea that the quantitative model is a good description of what happens in *S. pombe*.

Referee 3 states that the ‘results are interesting’, ‘are presented clearly and with appropriate controls’ and later says that it is a ‘nice paper’, but is not sure if it is sufficiently significant for publication in Nature.

Our lab has previously shown that a single G2/M CDK can drive the events of the cell cycle and that phosphorylation of substrates are ordered in the cell cycle by increasing CDK activity. In the present paper we show for the first time that a single G1/S CDK can also drive the events of the cell cycle, and importantly that *in vivo* G1/S and G2/M CDK activities are remarkably similar, with only a few qualitative differences. We also emphasise the importance of the centrosome in this regulatory system.

We believe that our proposal that CDK regulation of the cell cycle operating on a hybrid model which is predominantly quantitative with small qualitative features is of significant importance because it represents a different view from the general current consensus, whilst also being compatible with the majority of evidence gathered in other model organisms. Furthermore, we provide a methodology that allows our view to be tested in other eukaryotes.

Work on budding yeast has yielded a more nuanced picture. Although there are multiple cyclin-Cdk complexes activated in temporal waves, they go from least active to most active, in line with the quantitative model (Loog and Morgan Nature 2005; Kõivomägi et al. Mol Cell 2011). But the qualitative model pertains as well, and not just in a minor way. Notably, we now know that Cln3-Cdc28, the first of the cyclin-Cdc28 complexes, has a special ability to phosphorylate the C-terminal domain of Pol II at the initiation of the cell cycle (Kõivomägi et al. Science 2021). Likewise, an S-phase cyclin-Cdk complex (Clb5-Cdc28) phosphorylates some Cdk substrates (~25% of them), including known S-phase regulators, better than an M-phase cyclin-Cdk (Clb2-Cdc28) does (Loog and Morgan Nature 2005). So in budding yeast the general trend from low-to-high activity is there, but different cyclin-Cdk complexes are in fact specialized for different cell cycle events.

The referee raises examples from the budding yeast system that emphasise qualitative differences in substrate specificities between different CDKs as determined by *in vitro* protein kinase assays. However, this *in vitro* specialisation does not translate to essentiality for cell cycle progression, because no individual cyclin is essential in budding yeast. Indeed, similar to the fission yeast, only a single cyclin-CDK pair is necessary for the correct ordering of S-phase and chromosome separation, as shown recently by Pirincci-Ercan *et al.*⁴. These results indicate that any differences in substrate specificity between CDKs are not significant enough to prevent one cyclin-CDK complex from still being able to maintain temporal order of the cell cycle.

The referee mentions that they considered the Coudreuse & Nurse work, which showed the fission yeast cell cycle could be driven by a single cyclin-CDK (Cdc13-Cdc2), to 'convincingly argue for the quantitative model in fission yeast'. However, the report that the budding yeast nuclear division cycle of S-phase and mitosis can be driven by a single cyclin-CDK (Clb2-Cdc28) as mentioned above reports essentially the same observation as in the Coudreuse and Nurse paper⁴.

For the examples the referee provides, the majority of the characterisation of CDK substrate specificity is conducted *in vitro*. In contrast, our *in vivo* assays are more appropriate to understand the control system and global phosphorylation in the context of the cell. Furthermore, our paper provides a methodology that could be applied to budding yeast.

Finally, it should also be noted that the budding yeast cell cycle is unusual with respect to majority of other studied eukaryotes, as S-phase and mitotic events are partly overlapping (spindle formation and chromosome capture by microtubules occur in tandem with DNA replication), and budding is a specialised cell-cycle event. This may complicate interpretation and generalisation of results to the majority of other eukaryotes where S-phase and mitosis are temporally separated, and where cells do not divide by budding.

To address these points, we now describe in the discussion on lines 261-268 the evidence in budding yeast concerning qualitative *in vitro* differences between CDKs, but that even in this organism S-phase and mitosis can be driven by a single mitotic cyclin-CDK, although with delays. These observations align with and re-enforce our hybrid model of cell cycle progression.

In Xenopus extracts and mammalian cell lines, it is also clear that different cyclin-Cdk complexes are specialized for different cell cycle events, although the redundancy is greater than most people would have predicted a few decades ago. The Rb and E2F1 proteins are better substrates for cyclin A-Cdks than for cyclin B-Cdks even in vitro, because of cyclin-specific docking interactions (Schulman et al. PNAS 1998). Moreover and may more importantly, M-phase cyclin-Cdks (e.g. cyclin B-Cdk1) are unable to initiate DNA replication because unlike the S-phase cyclin-Cdks (cyclin E-Cdk2, cyclin A-Cdk2), they are excluded from the nucleus during interphase. Targeting cyclin B1-Cdk1 to the nucleus allows it to initiate DNA replication (Moore et al. Science 2003). Thus, there is a qualitative difference between the S-phase and M-phase cyclin-Cdk's localizations that makes the S phase cyclins good and the M phase cyclins bad at initiating S phase, and there are cyclin-specific docking interactions that reinforce that specificity.

So these are my main problems with the work: (1) I was already convinced that the quantitative model is the way to think of what happens in pombe; (2) I was already convinced that in budding yeast the quantitative model applies to an extent, but that there are also some major, not minor, specializations of individual cyclins for specific cell cycle events; and (3) It seems clear that in the

best-studied animal cell systems, qualitative differences in the cyclin-Cdks contribute in an important way to the ordering of cell cycle events. This is why I am not sure I support publication of this nice paper in Nature.

What we are proposing more generally for eukaryotic cell cycle control is that it is *predominantly* quantitative, which provides a very simple explanation for cell cycle temporal order, but there are also a small number of qualitative features. What is required now is a systematic approach to describing substrate specificities using *in vivo* protein kinase assays for G1/S and G2/M CDKs in other eukaryotes, as proposed in our paper.

Minor points:

1. How do the induced Cig2-Cdc2 and Cdc13-Cdc2 activities (Fig 2) compare to the activities present in the natural system?

In our experiments for Fig 2, we now outline in the text that cells continue to enter mitosis over the entire course of the experiment, thus making the entire range of cyclin-CDK expression physiologically relevant. We support this with additional experimental data in Extended data figure 4b, and outline this in the text [lines 106-109].

2. I do not mind speculation (e.g. lines 135-137) but I am not sure it is fair to say that “probably” the quantitative model applied to primordial eukaryotes.

The cyclins and CDKs are very likely to have arisen from gene duplication from a single ancestor given their sequence similarity. This implies that there was initially a single copy of both the cyclin and CDK component. Given that a qualitative model cannot apply to a situation in which a single cyclin and CDK are present, we believe our hypothesis that the quantitative model likely applied to a primordial eukaryote to be reasonable. We have now altered our statement to reflect that we mean before gene duplications [lines 140-141].

3. It is hard to distinguish between the two blue shades used in the plots in all of the main figures, particularly on the printed page. Perhaps black vs. red would be better.

We have now changed the colours of all main-text graphs.

Referee #4 (Remarks to the Author):

*The manuscript by Basu and colleagues utilized *S. pombe* to examine the effects of S-phase CDK and M-phase CDK activity on substrate phosphorylation. This is accomplished through a temperature-sensitive mutant that removes CDK activity at the restrictive temperature (36 degrees). The authors then add back in S-CDK and M-CDK fusion proteins that lack destruction boxes. The levels of the fusion proteins can be controlled by tetracycline. The authors' data demonstrates that most substrates equally phosphorylated by S-CDK or M-CDK. This reviewer was asked to specifically look at the mass-spectrometry based methods. The authors utilized a single 16plex TMT experiment in Figure 2 to examine the phosphorylation levels of peptides with increasing CDK activities. There are 8 expression (activity) levels for both the S-CDK and the M-CDK fusion proteins (8x2=16). The experiment setup is fundamentally sound. Since both CDK activities are always measured in the same 16plex there are no missing values in their dataset. The only issue with these data is that there are no replicates for measurements for each activation level—there is just the single 16plex. In theory, this isn't really a problem since there are lot of statistical tests for time-series data that do not use replicates. Overall, the authors cluster their 280 phosphorylation sites into several clusters and then show average membership with error bars. This is a good way to show the data for each cluster. The approach for the phosphorylation events is a good one with no significant issues within the dataset.*

Although the referee indicates the lack of replicates is not a problem, we have provided a biological repeat of the M-CDK induction to show and validate the reproducibility of our approach.

We present these results here as Figure R6. We do not wish to include these data as part of the current submission as they encompass other samples that are not relevant to this work, and are needed for other studies.

Figure R6. M-CDK Phosphoproteomics repeat.

- Heatmap of 210 detected CDK phosphorylation events that showed consistent phosphorylation behaviour and were present in both induction experiments. Sites are hierarchically clustered into 5 major clusters with more than 3 phosphosites (shown). 14 sites are not represented in these 5 clusters.
- Mean phosphorylation profiles of clusters shown in (a). Error bars give 95% CI. Where error bars are not shown, they are smaller than the size of the data point.
- Principal component analysis of the two M-CDK induction experiments in (a). Samples taken at the same timepoint are circled, and labelled with the time in minutes after M-CDK induction. Grey arrow gives direction over time.

Minor comment:

There are many grammatical and typographical errors throughout the manuscript.

We have now fixed the grammatical and typographical errors in the manuscript, and apologise that they were missed before.

References

- 1 Coudreuse, D. & Nurse, P. Driving the cell cycle with a minimal CDK control network. *Nature* **468**, 1074-1079, doi:10.1038/nature09543 (2010).
- 2 Patterson, J. O., Basu, S., Rees, P. & Nurse, P. CDK control pathways integrate cell size and ploidy information to control cell division. *Elife* **10**, doi:10.7554/eLife.64592 (2021).
- 3 Swaffer, M. P., Jones, A. W., Flynn, H. R., Snijders, A. P. & Nurse, P. CDK Substrate Phosphorylation and Ordering the Cell Cycle. *Cell* **167**, 1750-1761 e1716, doi:10.1016/j.cell.2016.11.034 (2016).
- 4 Pirincci Ercan, D. *et al.* Budding yeast relies on G1 cyclin specificity to couple cell cycle progression with morphogenetic development. *Sci Adv* **7**, doi:10.1126/sciadv.abg0007 (2021).

Reviewer Reports on the First Revision:

Referees' comments:

Referee #2 (Remarks to the Author):

In my original review, I suggested phosphomimicking substitution experiments, namely modifying the phosphosites that showed increased phosphorylation upon removal of centrosomal PP1. The previous version of the manuscript gave an impression that there are only very few such sites. The authors have now added data that the number of phosphosites is quite high, so the suggested experiment is not feasible.

I accept the authors' explanation and I have no further suggestions.

Referee #3 (Remarks to the Author):

I am satisfied with the authors' revisions.

Author Rebuttals to First Revision:

Referee #2 (Remarks to the Author):

In my original review, I suggested phosphomimicking substitution experiments, namely modifying the phosphosites that showed increased phosphorylation upon removal of centrosomal PP1. The previous version of the manuscript gave an impression that there are only very few such sites. The authors have now added data that the number of phosphosites is quite high, so the suggested experiment is not feasible.

I accept the authors' explanation and I have no further suggestions.

Referee #3 (Remarks to the Author):

I am satisfied with the authors' revisions.

We thank both of the Referees for their suggestions, our work was much improved for them.